# Statistical inference reveals the role of length, GC content, and local sequence in V(D)J nucleotide trimming

Magdalena L Russell[1,2]*, Noah Simon[3], Philip Bradley[1,4]*†,
Frederick A Matsen IV[1,5,6,7]*†

[1]Computational Biology Program, Fred Hutchinson Cancer Center, Seattle, United States; [2]Molecular and Cellular Biology Program, University of Washington, Seattle, United States; [3]Department of Biostatistics, University of Washington, Seattle, United States; [4]Institute for Protein Design, Department of Biochemistry, University of Washington, Seattle, United States; [5]Department of Genome Sciences, University of Washington, Seattle, United States; [6]Department of Statistics, University of Washington, Seattle, United States; [7]Howard Hughes Medical Institute, Seattle, United States

*For correspondence:
magruss@uw.edu (MLR);
pbradley@fredhutch.org (PB);
matsen@fredhutch.org (FAM)

†These authors contributed
equally to this work

Competing interest: The authors
declare that no competing
interests exist.

Reviewing Editor: Frederik
Graw, Friedrich-Alexander-
University Erlangen-Nürnberg,
Germany

**Abstract** To appropriately defend against a wide array of pathogens, humans somatically generate highly diverse repertoires of B cell and T cell receptors (BCRs and TCRs) through a random process called V(D)J recombination. Receptor diversity is achieved during this process through both the combinatorial assembly of V(D)J-genes and the junctional deletion and insertion of nucleotides. While the Artemis protein is often regarded as the main nuclease involved in V(D)J recombination, the exact mechanism of nucleotide trimming is not understood. Using a previously published TCRβ repertoire sequencing data set, we have designed a flexible probabilistic model of nucleotide trimming that allows us to explore various mechanistically interpretable sequence-level features. We show that local sequence context, length, and GC nucleotide content in both directions of the wider sequence, together, can most accurately predict the trimming probabilities of a given V-gene sequence. Because GC nucleotide content is predictive of sequence-breathing, this model provides quantitative statistical evidence regarding the extent to which double-stranded DNA may need to be able to breathe for trimming to occur. We also see evidence of a sequence motif that appears to get preferentially trimmed, independent of GC-content-related effects. Further, we find that the inferred coefficients from this model provide accurate prediction for V- and J-gene sequences from other adaptive immune receptor loci. These results refine our understanding of how the Artemis nuclease may function to trim nucleotides during V(D)J recombination and provide another step toward understanding how V(D)J recombination generates diverse receptors and supports a powerful, unique immune response in healthy humans.

## Editor's evaluation

Russell et al. study and reveal compelling evidence for potential sequence-based factors that may drive VDJ trimming, a mechanism involved in VDJ recombination that shapes adaptive immune repertoire generation. The work is based on a rigorous statistical comparison of logistic regression models to reveal the role and function of cutting enzymes in shaping T- and B-cell receptor diversity which could provide fundamental new insights into these processes.

## Introduction

Cells throughout the body regularly present protein fragments, known as antigens, on cell-surface molecules called major histocompatibility complex (MHC). Receptors on the surface of T cells can bind to these MHC-bound antigens, recognize them, and, if necessary, initiate an immune response. For an individual to be capable of defending against a wide array of potential foreign pathogens, they somatically generate a massive diversity of T cell receptors (TCRs) through a random process called V(D)J recombination. After generation, TCRs undergo a selection process to ensure proper expression, MHC recognition, and limited autoreactivity. The collection of TCRs in an individual comprises their TCR repertoire.

The majority of human T cells express α-β receptors that consist of an α and a β protein chain. During the V(D)J recombination process of the β chain, a single V-, D-, and J-gene are randomly chosen from a pool of V-gene, D-gene, and J-gene segments within the germline TCRβ locus over a series of steps. To begin this process, the recombination activating gene protein complex aligns a randomly chosen D- and J-gene, removes the intervening chromosomal DNA between the two genes, and forms a hairpin loop at the end of each gene (*Gellert, 1994*; *Fugmann et al., 2000*; *Schatz and Swanson, 2011*). Each hairpin loop is then nicked open, typically in an asymmetrical fashion, by the Artemis:DNA-PKcs protein complex (*Ma et al., 2002*; *Lu et al., 2007*). This asymmetrical hairpin opening creates a single-stranded DNA overhang at the end of both genes that, depending on the location of the hairpin nick, may contain P-nucleotides (short palindromes of gene terminal sequence) (*Gauss and Lieber, 1996*; *Nadel and Feeney, 1997*; *Ma et al., 2002*; *Jackson et al., 2004*; *Lu et al., 2007*). The most dominant hairpin opening position leads to a single-stranded 3' overhang that is 4 nucleotides in length (2 nucleotides of which are P-nucleotides) (*Lu et al., 2007*). From here, nucleotides may be deleted from each gene end through an incompletely understood mechanism suggested to involve Artemis (*Feeney et al., 1994*; *Nadel and Feeney, 1995*; *Nadel and Feeney, 1997*; *Jackson et al., 2004*; *Gu et al., 2010*; *Chang et al., 2015*; *Chang and Lieber, 2016*; *Zhao et al., 2020*; *Russell et al., 2022b*). This nucleotide trimming can remove traces of P-nucleotides (*Gauss and Lieber, 1996*; *Srivastava and Robins, 2012*). Non-template-encoded nucleotides, known as N-insertions, can also be added to each gene end by the enzyme terminal deoxynucleotidyl transferase (*Kallenbach et al., 1992*; *Gilfillan et al., 1993*; *Komori et al., 1993*). Once the nucleotide addition and deletion steps are completed, the gene segments are paired and ligated together (*Zhao et al., 2020*). From here, the process is repeated between a random V-gene and this combined D-J junction to complete the TCRβ chain. A similar TCR chain recombination then proceeds, though without a D-gene, to complete the α-β TCR. Other adaptive immune receptor loci, such as *TRG*, *TRD*, and all *IG* loci, also undergo V(D)J recombination during the development of γ-δ T cells and B cells, respectively.

Junctional diversity created by the deletion and non-templated insertion of nucleotides during V(D)J recombination contributes substantially to the resulting diversity of the TCR repertoire. Small variations in gene sequence have been shown to lead to large changes in the extent of nucleotide deletion (*Nadel and Feeney, 1995*; *Gauss and Lieber, 1996*; *Nadel and Feeney, 1997*; *Jackson et al., 2004*). For example, sequences with high AT content suffer greater nucleotide loss than sequences with high GC content (*Gauss and Lieber, 1996*). These findings are suggestive of a nuclease that either binds an AT-rich sequence motif or requires an AT-specific structure (e.g. a sequence that breathes, *Tsai et al., 2009*), however, further work is required to quantify this mechanistic preference.

The Artemis protein is often regarded as the main nuclease involved in V(D)J recombination (*Chang et al., 2015*; *Chang and Lieber, 2016*; *Zhao et al., 2020*). Artemis is a member of the metallo-$\beta$-lactamase family of nucleases (*Moshous et al., 2001*) and is widely regarded as a structure-specific nuclease as opposed to a nuclease that binds specific DNA sequences (*Ma et al., 2005*; *Chang et al., 2015*; *Chang and Lieber, 2016*; *Yosaatmadja et al., 2021*). Members of this family are characterized by their conserved metallo-β-lactamase and β-CASP domains and their ability to nick DNA or RNA in various configurations (*Dominski, 2007*; *Pettinati et al., 2016*). Alone, Artemis possesses intrinsic 5'-to-3' exonuclease activity on single-stranded DNA (*Li et al., 2014*). On double-stranded DNA, Artemis, in complex with DNA-PKcs, has endonuclease activity on 5' and 3' DNA overhangs and on DNA hairpins (*Ma et al., 2002*; *Lu et al., 2007*; *Lu et al., 2008*). It has been proposed that the Artemis:DNA-PKcs complex binds single-stranded-to-double-stranded DNA boundaries prior to nicking (*Ma et al., 2002*; *Ma et al., 2005*; *Lu et al., 2007*; *Chang and Lieber, 2016*); for blunt DNA ends, previous work has concluded that sequence-breathing is required to achieve this

structural configuration prior to Artemis action (*Chang et al., 2015*). Further, Artemis, in complex with XRCC4-DNA ligase IV, has additional endonuclease activity on 3' DNA overhangs and preferentially nicks one nucleotide at a time from the single-stranded 3' end (*Chang et al., 2016*; *Gerodimos et al., 2017*). Despite these diverse nucleolytic functions, the extent of involvement and exact mechanism of action for the Artemis protein during the nucleotide trimming step of V(D)J recombination, and how it relates to observed sequence-dependent changes in trimming (*Nadel and Feeney, 1995*; *Gauss and Lieber, 1996*; *Nadel and Feeney, 1997*; *Jackson et al., 2004*), has yet to be fully understood.

While molecular experiments using model organisms have been essential for establishing the current mechanistic understanding of the nucleotide trimming process, studies in humans have been limited. Statistical inference on high-throughput repertoire sequencing data sets allows for exploration of the in vivo V(D)J recombination mechanism outside of model organisms. In particular, analysis of trimming in high-throughput data sets should lead to insights about the natural underlying process, in the same way that analysis of large data sets has led to insight into the process of somatic hypermutation. There, researchers have found quite significant connections between local sequence identity and mutation patterns, leading to a rich literature (*Rogozin and Kolchanov, 1992*; *Dunn-Walters et al., 1998*; *Cohen et al., 2011*; *Yaari et al., 2013*; *Elhanati et al., 2015*; *Wei et al., 2015*; *Cui et al., 2016*; *Feng et al., 2019*; *Spisak et al., 2020*).

In contrast, we are only aware of one statistical analysis connecting sequence identity to trimming lengths (*Murugan et al., 2012*). This one existing analysis (*Murugan et al., 2012*) has shown that a simple position-weight-matrix style (PWM) model does a surprisingly good job of predicting the distribution of trimming lengths for a variety of V-genes. However, while this trimming model has good model fit and predictive accuracy, it is limited by the assumption that the trimming mechanism relies solely on a sequence motif and, as such, is not designed in a way that allows us to explore alternative hypotheses.

In this paper, we explore the sequence-level determinants of nucleotide trimming during V(D)J recombination using statistical inference on high-throughput TCRβ repertoire sequencing data (*Emerson et al., 2017*). With the goal of informing our mechanistic understanding in a quantitative way, we have designed a flexible probabilistic model of nucleotide trimming that allows us to explore various sequence-level features. Our results show that trimming probabilities are highest for DNA positions near the end of the sequence that contain high GC content upstream, quantifying the role of sequence-breathing dynamics in the trimming process. We also see evidence of a sequence motif that appears to get preferentially trimmed, independent of possible sequence-breathing effects. As such, we can predict trimming probabilities most accurately using a model that includes features for local sequence context, length, and GC nucleotide content in both directions of the wider sequence. We show that this model has high predictive accuracy for V- and J-gene sequences from an independent TCRβ-sequencing data set, and also extends well to TCRα, TCRγ, and IGH sequences. Further, we demonstrate that genetic variations within the gene encoding the Artemis protein that were previously identified as being associated with increasing the extent of trimming (*Russell et al., 2022b*) are also associated with changes in several model coefficients.

## Results
### Training data description

We worked with TCRβ-immunosequencing data representing 666 individuals (*Emerson et al., 2017*). V(D)J recombination scenarios were assigned to each sequence from each individual using the IGoR software which is designed to learn unbiased V(D)J recombination statistics from immune sequence reads (*Marcou et al., 2018*). Using these V(D)J recombination statistics, IGoR output a list of potential recombination scenarios with their corresponding likelihoods for each TCRβ-chain sequence in the training data set. We annotated each sequence with a single V(D)J recombination scenario by sampling from these potential scenarios according to the posterior probability of each scenario (see Materials and methods for further details).

Annotated TCR sequences can be separated into two categories: 'productive' rearrangements which code for a complete, full-length protein and 'non-productive' rearrangements which do not. Non-productive sequences are generated when the V(D)J recombination process produces a sequence that is either out-of-frame or contains a stop codon. Each T cell contains two loci which can

undergo the V(D)J recombination process. When the first recombination fails to generate a functional receptor (creating a non-productive sequence), followed by a successful rearrangement on the T cell's second chromosome (a productive sequence), the non-productive rearrangement can be sequenced as part of the repertoire. Non-productive sequences do not generate proteins that undergo functional selection in the thymus, and their recombination statistics should reflect only the V(D)J recombination generation process (*Robins et al., 2010*; *Murugan et al., 2012*; *Sethna et al., 2019*). In contrast, the recombination statistics of productive sequences should reflect both V(D)J recombination generation and functional selection. Because we are interested in nucleotide trimming during the V(D)J recombination generation process, prior to selection, we only include non-productive sequences in our training data set. Further, because V-gene sequences within the *TRB* locus contain more sequence variation than D- and/or J-genes, we only include V-gene sequences in our training data set.

## Replicating a previous model of nucleotide trimming

The extent of nucleotide trimming varies substantially from gene to gene (*Nadel and Feeney, 1995*; *Nadel and Feeney, 1997*; *Jackson et al., 2004*; *Murugan et al., 2012*). Previous work has identified an interesting impact of sequence features, such as sequence nucleotide context, on trimming probabilities using a PWM model (*Murugan et al., 2012*). To our knowledge, this is the only model that takes nucleotide sequence identity into account when predicting trimming probabilities. Specifically, this model leverages a 'trimming motif' containing 2 nucleotides 5' of the trimming site and 4 nucleotides 3' of the trimming site to predict the probability of trimming at a given site. It was designed and trained using sequencing data from just nine individuals (*Murugan et al., 2012*), and has surprisingly good model fit and predictive accuracy across many V-genes despite its simplicity. Using a different, and much larger, repertoire sequencing data set, we have trained this PWM model and replicated previous work (*Figure 2—figure supplement 1*). We will refer to this model as the *2×4 motif* model. It is important to note that this PWM model is not the primary model described in *Murugan et al., 2012*, but again is the only one that relates nucleotide identity to trimming.

## Model set-up overview

While the *2×4 motif* model has good predictive accuracy and model fit (*Murugan et al., 2012*), it is limited by its assumption that the trimming mechanism relies solely on a sequence motif. Here, we have generalized this PWM model to a model that allows for arbitrary sequence features, and trained each new model using conditional logistic regression (see Materials and methods). With this set-up, we were able to evaluate the relative importance of new mechanistically interpretable features for predicting trimming probabilities. Specifically, we designed features to measure the effects of DNA-shape, length, and GC nucleotide content in both directions of the wider sequence on the probability of trimming at a given position in a gene sequence.

We parameterize each of these features as follows. An example of how an arbitrary V-gene sequence is transformed into features for modeling is shown in *Figure 1*. To parameterize DNA-shape, we used previously developed methods (*Zhou et al., 2013*; *Chiu et al., 2016*) to estimate various DNA-shape values (i.e. roll, twist, electrostatic potential, minor groove width, etc.) for each single-nucleotide position within a sequence window surrounding the trimming site. To parameterize length, we measure the sequence-independent distance from the end of the gene (i.e. the number of nucleotides from the 3'-end of the sequence) as an integer-valued variable. We parameterize GC nucleotide content using the raw counts of AT and GC nucleotides on both sides of the trimming site (the *two-side base-count*). By using raw nucleotide counts, this measure also serves to parameterize length. Because AT nucleotides have a greater potential for sequence-breathing compared to GC nucleotides within a sequence (*Jose et al., 2009*), these *two-side base-count* terms may be serving as a proxy for the capacity of a sequence to breathe. As such, because sequence-breathing potential is only relevant for nucleotides that are paired, we do not include the nucleotides within the 3' single-stranded-overhang when counting 3' AT and GC nucleotides (see Appendix 2).

With these features, we designed models containing various feature combinations (*Figure 1B*). Collectively, these models allow us to explore other possible sequence-level determinants of nucleotide trimming, in addition to the previously proposed (*Murugan et al., 2012*) "trimming motif" hypothesis. We trained each model using the V-gene training data set described above (see Materials and methods for further model training details), and evaluated performance using a suite of different

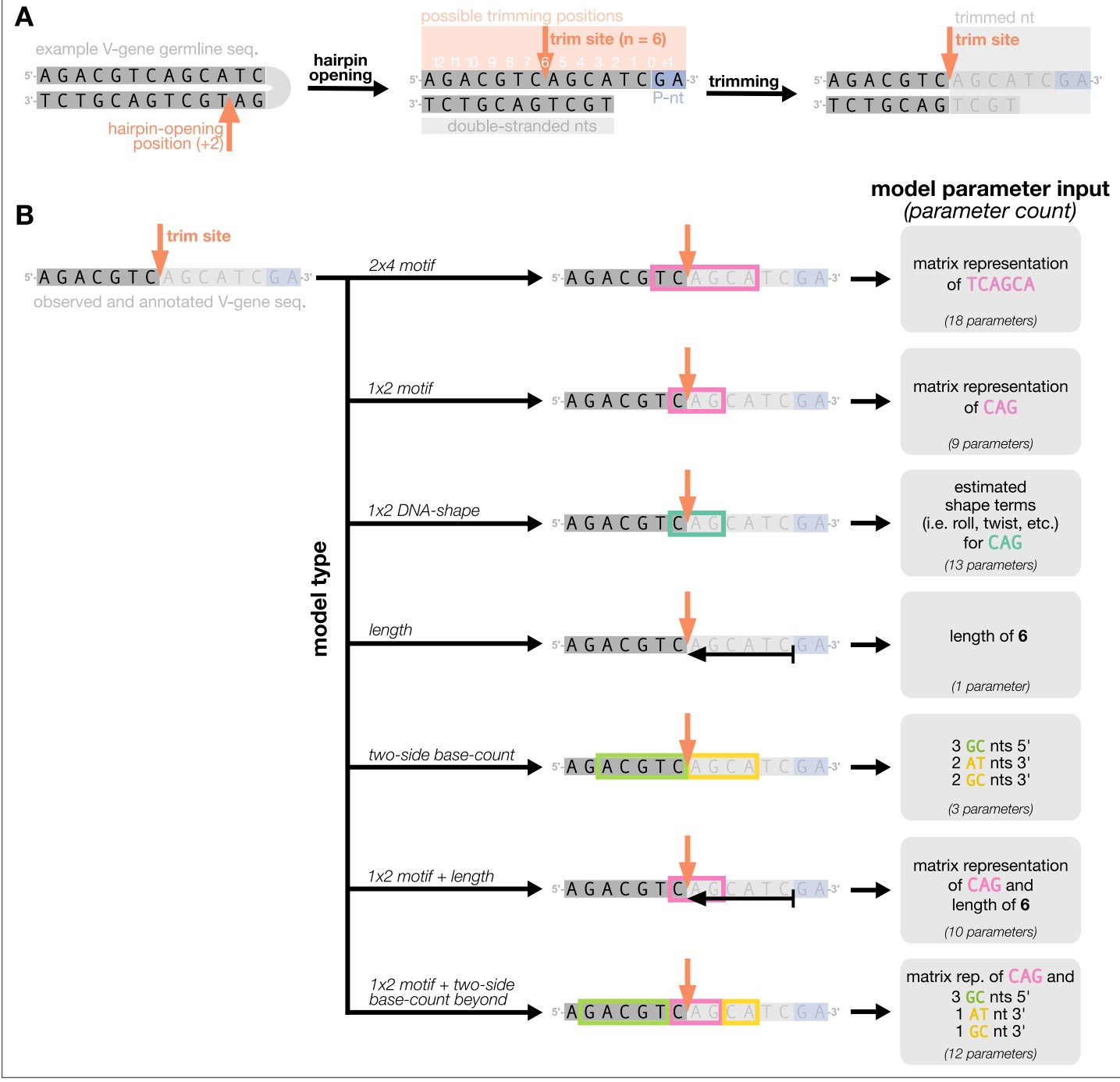

**Figure 1.** Overview of how a sequence is transformed into features for regression. (**A**) As described, during the early stages of V(D)J recombination between two genes, the hairpin of each gene is opened; here, we are showing this hairpin-opening step for a single arbitrary V-gene. The most common hairpin-opening position leads to a 4-nucleotide-long single-stranded overhang (2 nucleotides of which are considered P-nucleotides, as shown in purple). From here, each gene can undergo nucleotide trimming. In this example, the V-gene is trimmed back 6 nucleotides. (**B**) All models were trained with non-productive V-gene sequences whose trimming positions were inferred during a sequence annotation step. For our model parameterization, we only consider the top strand (5'-to-3') of the observed sequence. Here, the sequence features parameterized for each model type are shown for the example sequence from (**A**). The pink boxes surround nucleotides included in the matrix representation of *motif* features. The turquoise boxes surround nucleotides used to estimate and parameterize *DNA-shape* features (see Appendix 2 for further details). The green boxes surround nucleotides included in the counts of GC nucleotides 5' of the trimming site; in our actual models, we count nucleotides within a 10-nucleotide window (a 5-nucleotide window is shown in the figure). Because this window size is fixed, we do not need to include an additional parameter for AT nucleotide count 5' of the trimming site (since it is already indirectly modeled). The yellow boxes surround double-stranded nucleotides included in the

*Figure 1 continued on next page*

*Figure 1 continued*

counts of AT and GC nucleotides 3' of the trimming site. These raw 3'-nucleotide counts also indirectly parameterize length; as such, we never include both *length* and *two-side base-count* parameters in the same model. In addition to the models shown in the figure, we also evaluated a *null* model which does not contain any parameters.

held-out data groups (*Figure 2*). Specifically, to evaluate model fit, we computed the expected per-sequence conditional log loss of each model using the full V-gene training data set.

To evaluate model generalizability, we computed the expected per-sequence conditional log loss using the following held-out groups:

- many random, held-out subsets of the V-gene training data set;
- held-out subsets of the V-gene training data set containing groups of V-genes defined to be the 'most-different' from all other genes using either the terminal sequences (last 25 nucleotides of each sequence) or the full gene sequences;
- the full J-gene data set.

For each of these held-out group analyses, each model was re-trained using the full V-gene training data set with the held-out group-of-interest removed (see Materials and methods and Appendix 3 for

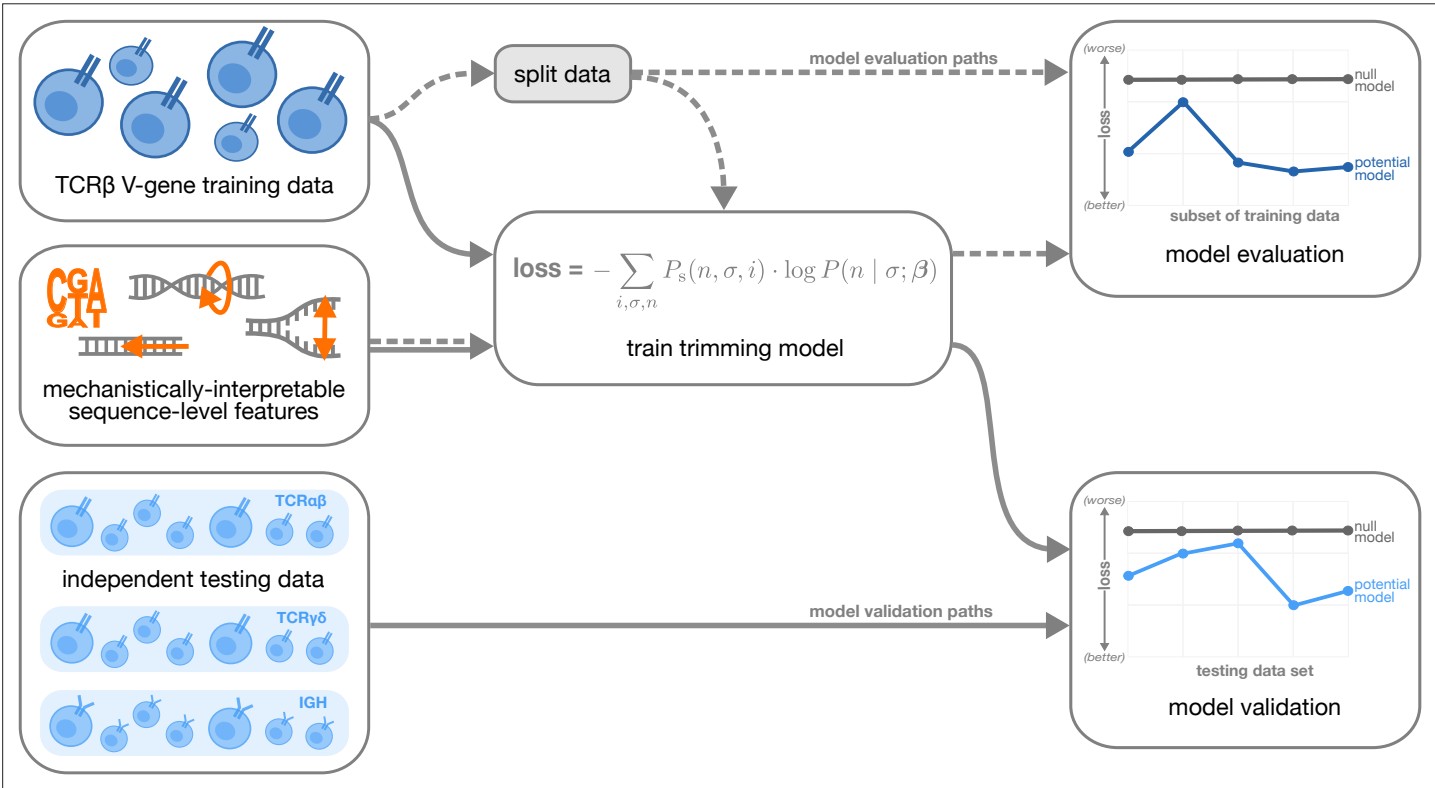

**Figure 2.** Overview of analysis strategy. The T cell receptor (TCR)β V-gene training data was used to train each trimming model containing various combinations of sequence-level features (*Figure 1*) by minimizing the associated loss function. The loss function is given by a sum across individuals $i$, genes $\sigma$, and trimming lengths $n$ of the sampling probability of each observation $P_s$ multiplied by the gene-specific trimming probability predicted by a model with β parameters (see Materials and methods for further details). Each potential model first underwent a 'model evaluation' stage (shown by the dashed lines) during which the model performance was evaluated using various subsets of the training TCRβ V-gene data set. Once all models were evaluated, a subset of the potential models continued on to the 'model validation' stage (shown by the solid lines) during which the performance of the model coefficients from the previous TCRβ V-gene training run were validated using several independent testing data sets including TCRβ, TCRα, TCRγ, and IGH sequences. At each stage, the performance of each model was compared to a null model (containing zero parameters, see Materials and methods).

The online version of this article includes the following figure supplement(s) for figure 2:

**Figure supplement 1.** Using a different, and much larger, repertoire sequencing data set, we have closely replicated previous work (*Murugan et al., 2012*) which illustrated that a simple position-weight-matrix-style model has good predictive accuracy for many T cell receptor β V-genes.

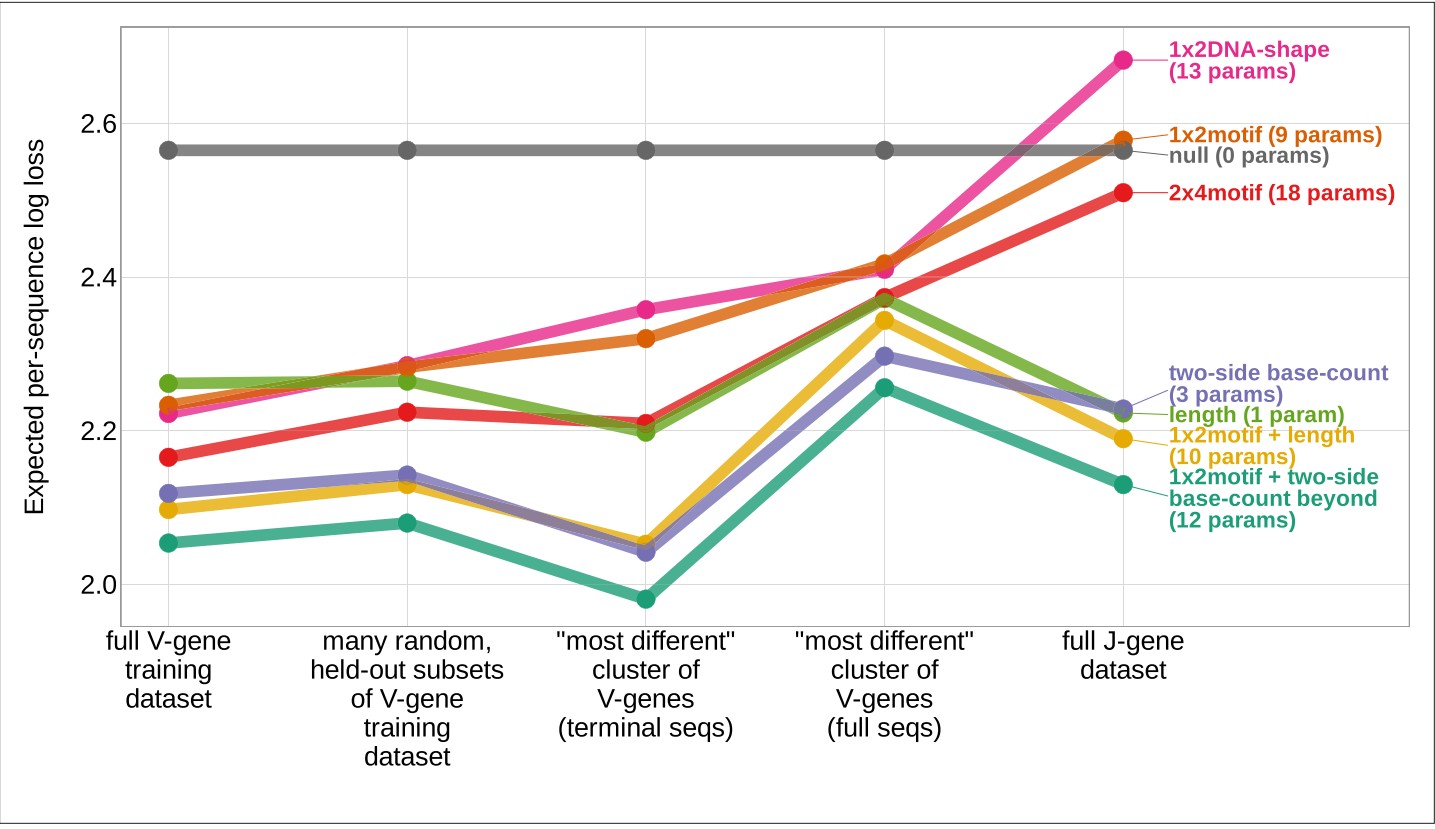

**Figure 3.** Expected per-sequence conditional log loss computed for various models using the full V-gene training data set, many random, held-out subsets of the V-gene training data set, a held-out subset of the V-gene training data set containing a group of V-genes defined to be the 'most-different' using the terminal sequences (last 25 nucleotides of each sequence), a held-out subset of the V-gene training data set containing a group of V-genes defined to be the 'most-different' using the full gene sequences, and the full J-gene data set. Each model was trained using the full V-gene training data set with the held-out group or 'most-different' group (if applicable) removed (see Materials and methods and Appendix 3). Lower expected per-sequence log loss corresponds to better a model fit. The *1×2 motif + two-side base-count beyond* model has the best model fit and generalizability across all data sets.

The online version of this article includes the following source data and figure supplement(s) for figure 3:

**Source data 1.** Expected per-sequence conditional log loss reported for each model and validation data set.

**Figure supplement 1.** For each model, there was some variation in the expected per-sequence conditional log loss values computed across the 20 random, held-out subsets of the V-gene training data set.

further details) prior to computing the loss. A lower expected per-sequence conditional log loss indicated better model fit and/or model generalizability. Following this model evaluation, we validated a subset of the models by using the model coefficients from the previous TCRβ V-gene training run and computing the expected per-sequence conditional log loss of the model using several independent testing data sets (*Figure 2*).

## Local sequence context, length, and GC nucleotide content in both directions of the wider sequence, together, accurately predict the trimming probabilities of a given V-gene sequence

In an effort to capture the complex underlying biochemistry of the deletion process, we trained models containing various combinations of sequence-level feature types (*Figure 1B*) and evaluated their ability to accurately predict V-gene trimming probabilities. With this approach, we found that a model containing parameterizations of local sequence context, length, and GC nucleotide content in both directions of the wider sequence (the *1×2 motif + two-side base-count beyond* model) had the best model fit and generalizability across different data sets (*Figure 3* and *Figure 3—figure supplement 1*). This model contains a *1×2 motif*, including 1-nucleotide position 5′ of the trimming

site and 2-nucleotide positions 3' of the trimming site within the trimming window, and includes only bases beyond this trimming window in the AT and GC *two-side base-count* terms (*Figure 1*). Despite containing fewer total parameters than the original *2×4 motif* model (*Murugan et al., 2012*) (12 parameters compared to 18 parameters), the *1×2 motif + two-side base-count beyond* model had better predictive accuracy (*Figure 4* and *Figure 4—figure supplement 1*).

We considered the significance of the inferred model coefficients using a Bonferroni-corrected significance threshold of 0.0033 (corrected for the total number of model coefficients). With this threshold, we found that many of the inferred model coefficients were significant and quantified mechanistic patterns. Each coefficient represents the change in log10 odds of trimming at a given site resulting from an increase in the feature value, given that all other features are held constant. Within the nucleotides immediately surrounding the trimming site, bases 5' of the trimming site have a slightly stronger effect on the trimming probability than bases 3' of the trimming site (*Figure 4B*). Specifically, 5' of the trimming site, C nucleotides have a strong positive effect on the trimming probability ($\log_{10}$ coefficient = 0.2388) whereas A and T nucleotides have a negative effect ($\log_{10}$ coefficient$_A$ = −0.108 and $\log_{10}$ coefficient$_T$ = −0.137). In contrast, immediately 3' of the trimming site, G and T nucleotides have a positive effect on the trimming probability ($\log_{10}$ coefficient$_G$ = 0.093 and $\log_{10}$ coefficient$_T$ = 0.125) whereas C nucleotides have a negative effect ($\log_{10}$ coefficient = −0.174). These results suggest a different possible mechanistic pattern than previous *motif*-only models (*Murugan et al., 2012*; *Figure 2—figure supplement 1B*). Further, beyond the *1×2 motif* sequence window, the count of GC nucleotides 5' of the motif (within a 10-nucleotide window) has a strong positive effect on the trimming probability ($\log_{10}$ coefficient = 0.164) (*Figure 4C*). The counts of both AT and GC nucleotides 3' of the motif have a strong negative effect on the trimming probability ($\log_{10}$ coefficient$_{AT}$ = −0.123 and $\log_{10}$ coefficient$_{GC}$ = −0.126). Interestingly, the magnitude of these negative effects are very similar between AT and GC counts. This suggests that the raw number of nucleotides 3' of the motif (e.g. the length) is more important for predicting the trimming probability at a given site compared to the identity of the nucleotides. p-values for each of these coefficients were reported to be smaller than machine tolerance ($2.23 \times 10^{-308}$). We noted minimal variation in the magnitude of each inferred coefficient even when changing the number of sequences included in the training data set (*Figure 4—figure supplement 7*).

Because we were interested in parameterizing sequence-breathing effects using the *two-side base-count* terms, we only included nucleotides that are considered to be double-stranded after hairpin-opening within each count. In our modeling, we assume that the DNA hairpin is opened at the +2 position, leading to a 4-nucleotide-long 3'-single-stranded-overhang (the 2 nucleotides furthest 3' are considered P-nucleotides) (*Ma et al., 2002*; *Lu et al., 2007*). As such, the first 2 nucleotides of the gene sequence can be considered single-stranded, and we do not include them in the *two-side base-count* terms. When we train a model that ignores this distinction, and include all gene sequence nucleotides in the *two-side base-count* terms, we note very similar inferred coefficients and model fit (*Figure 4—figure supplement 2*). We acknowledge that other hairpin-opening positions may be possible. To explore whether the +2-hairpin-opening-position assumption could be affecting our inferences, we trained the *1×2 motif + two-side base-count beyond* model with other possible hairpin-opening-position assumptions and noted minimal variation in model fit (*Figure 4—figure supplement 3*).

We also evaluated the predictive accuracy of *motif + two-side base-count beyond* models containing different 'trimming motif' sizes. We find that models containing a small motif (e.g. a *1×2 motif*) achieve similar predictive accuracy and are more generalizable compared to models containing a larger motif (*Figure 4—figure supplement 4*).

Because the trimming mechanism is thought to be consistent across V-, D-, and J-genes from both productive and non-productive sequences, we were also interested in whether the inferred coefficients for the *1×2 motif + two-side base-count beyond* model would be consistent between the model trained using the non-productive V-gene training data set, a model trained using a non-productive J-gene data set, and a model trained using a productive V-gene data set. As such, we trained a new *1×2 motif + two-side base-count beyond* model using only non-productive J-gene sequences and a separate, new *1×2 motif + two-side base-count beyond* model using only productive V-gene sequences (both sequence sets were from the same cohort of individuals as the V-gene training data set). We found that the inferred coefficients were highly similar between the three models (*Figure 4—figure supplement 5* and *Figure 4—figure supplement 6*).

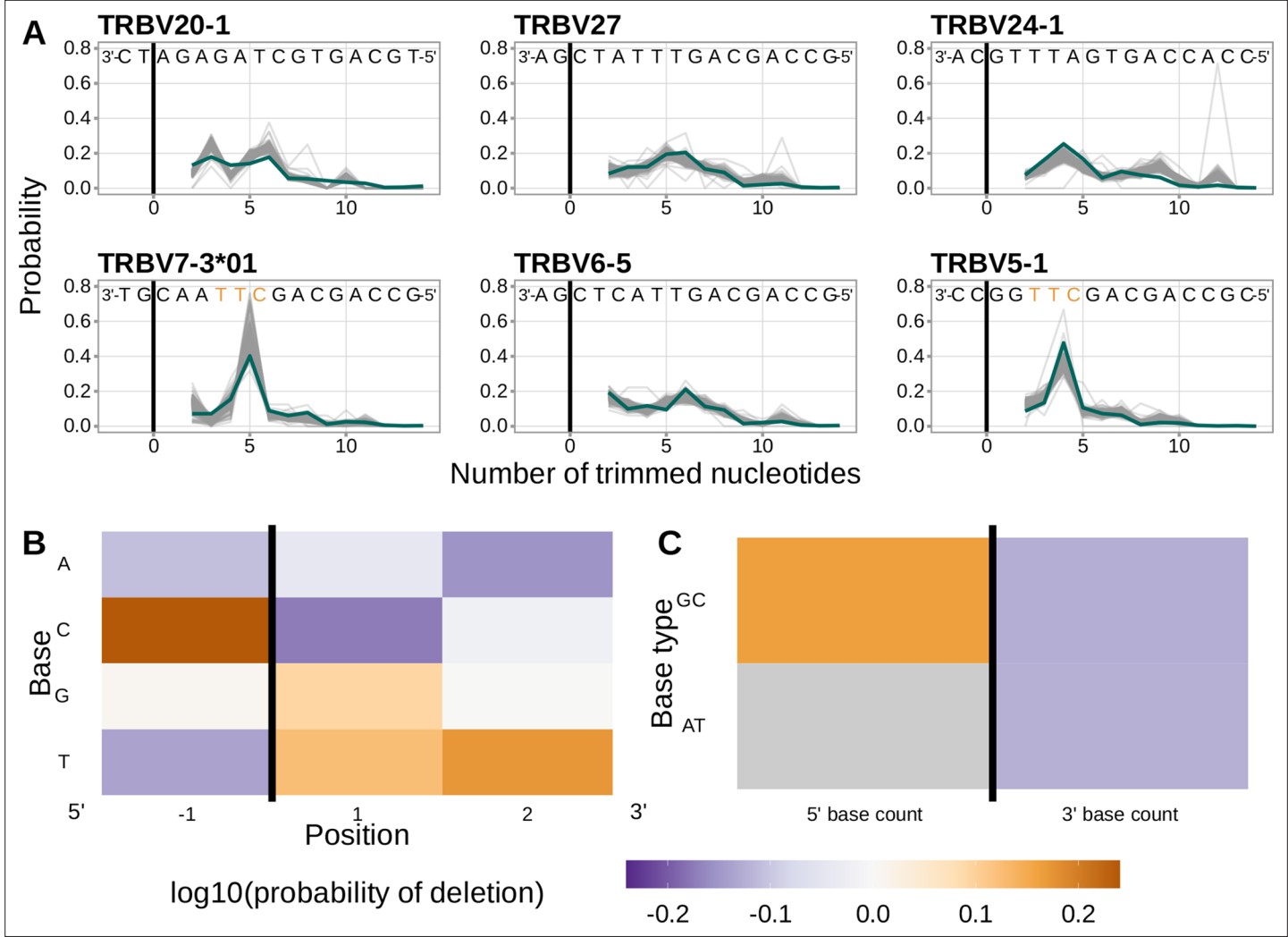

**Figure 4.** Performance of the *1×2 motif + two-side base-count beyond* model. (**A**) Inferred trimming profiles using the *1×2 motif + two-side base-count beyond* model have good predictive accuracy overall; here, we show the inferred trimming profiles (in blue) for the most frequently used V-genes. Gene-specific trimming profiles for each individual in the training data set are shown in gray. The sequence context with the highest probability of trimming (3'-TTC-5' or 3'-TGC-5') from (**B and C**) is highlighted in orange. (**B**) Position-weight-matrix of the local sequence context dependence of V-gene trimming probabilities consisting of 1 nucleotide 5' of the trimming site and 2 nucleotide 3' of the trimming site from fitting the *1×2 motif + two-side base-count beyond* model. Positions 5' and 3' of the trimming site have a strong effect on the probability of trimming. (**C**) Inferred *two-side base-count beyond* model coefficients from fitting the *1×2 motif + two-side base-count beyond* model suggest that the count of GC bases 5' of the motif has a strong positive effect on the trimming probability whereas the count of GC and/or AT bases 3' of the motif has a negative effect. The count of AT nucleotides 5' of the motif (shown in gray) was not included in this model. The black vertical line corresponds to the trimming site. Each inferred coefficient is given as the change in log10 odds of trimming at a given site resulting from an increase in the feature value, given that all other features are held constant.

The online version of this article includes the following source data and figure supplement(s) for figure 4:

**Source data 1.** Inferred and observed trimming profiles for the most frequently used V-genes in the V-gene training data set.

**Source data 2.** Inferred 1x2 motif + two-side base-count beyond model coefficients.

**Figure supplement 1.** Performance of the *1×2 motif + two-side base-count beyond* model across all TRB V-genes, ordered by the frequency of usage in the training data set.

**Figure supplement 2.** Including all gene sequence nucleotides in the *two-side base-count* terms, instead of restricting to double-stranded nucleotides, leads to very similar inferred coefficients and model fit.

**Figure supplement 3.** The assumed position of the initial hairpin-opening nick during the early stages of V(D)J recombination has little effect on the inferred coefficients and model fit.

*Figure 4 continued on next page*

*Figure 4 continued*

**Figure supplement 4.** Models containing a small motif (e.g.a *1×2 motif*) achieve similar predictive accuracy and are more generalizable compared to models containing a larger motif.

**Figure supplement 5.** Inferred coefficients from a *1×2 motif + two-side base-count beyond* model trained using only J-gene sequences are highly similar to those from the model trained using the V-gene training data set.

**Figure supplement 6.** Inferred coefficients from a *1×2 motif + two-side base-count beyond* model trained using only productive V-gene sequences are highly similar to those from the model trained using the non-productive V-gene training data set.

**Figure supplement 7.** The magnitudes of the inferred coefficients from the *1×2 motif + two-side base-count beyond* model have minimal variance when changing the number of sequences included in the training data set.

When evaluating models containing only a single feature type, we find that the *two-side base-count* model which parameterizes GC nucleotide content on both sides of the trimming site (and, indirectly, length) has the best model fit and generalizability across all held-out groups tested (*Figure 3*). As such, these GC-content features, which are likely parameterizing the capacity for the sequence to breathe, are more predictive of V-gene trimming probabilities than local sequence context or DNA-shape alone. This finding supports previous observations that Artemis may act as a structure-specific nuclease as opposed to a nuclease that binds specific DNA sequences (*Ma et al., 2005*; *Chang et al., 2015*; *Chang and Lieber, 2016*; *Yosaatmadja et al., 2021*).

## Inferred local sequence context coefficients suggest a biological trimming motif

A persistent concern with the *1×2 motif + two-side base-count beyond* model was that the *motif* coefficients could be driven by certain genes, instead of representing an actual gene-segment-wide signal. When comparing the inferred trimming profiles from the *two-side base-count* model to those from the *1×2 motif + two-side base-count beyond* model, we identified a group of V-genes which had drastically lower prediction error when the *1×2 motif* terms were included. These V-genes had a difference in per-gene root mean squared error between the two models that was greater than –0.13 (*Figure 5A*). The genes included in this group were *TRBV5-3*, *TRBV7-3\*01*, *TRBV7-3\*04*, *TRBV7-4*, *TRBV9*, *TRBV11*, and *TRBV13*. To evaluate whether these genes could be driving the observed *motif* signal, we explored whether the prediction error for these genes changed when they were removed from the model training data set.

In fact, we found that the inferred trimming profiles for these genes still had very low prediction error despite the genes not being included in the model training data set (*Figure 5B and C*), showing the generalizability of these features. The inferred model coefficients from this *1×2 motif + two-side base-count beyond* model fit using the subsetted training data set were highly similar to those from the original model fit using the full training data set. Because genes which are highly similar sequence-wise to the group of held-out genes could still be present in the training data set and be driving these similarities, we defined a new data set that excluded this larger group of genes. When we repeated the same experiment with this new, more-restricted training data set, we observed similar results (*Figure 5B and C*). As such, both of these experiments provided evidence that the *motif* signal may actually represent a gene-segment-wide sequence motif that appears to get preferentially trimmed, independent of GC-content-related effects.

## Trimming-associated variation within the Artemis locus is associated with a change in model coefficients

Using a subset of the V-gene training data set used here, we previously identified a set of single nucleotide polymorphisms (SNPs) within the gene encoding the Artemis protein that are associated with increasing the extent of V- and J-gene trimming (*Russell et al., 2022b*). This result suggested that trimming profiles may subtly vary in the context of these SNPs. As such, we were interested in whether these SNPs could be mediating (or serving as a proxy for) a change in the trimming mechanism. To explore this, we worked with paired SNP array and TCRβ-immunosequencing data representing 611 of the original 666 individuals in the V-gene training data set used here. Our previous work *Russell et al., 2022b* used data from only 398 of these individuals, however, the conclusions of that paper held when using this expanded group of 611 individuals in the analysis. With these data, we asked

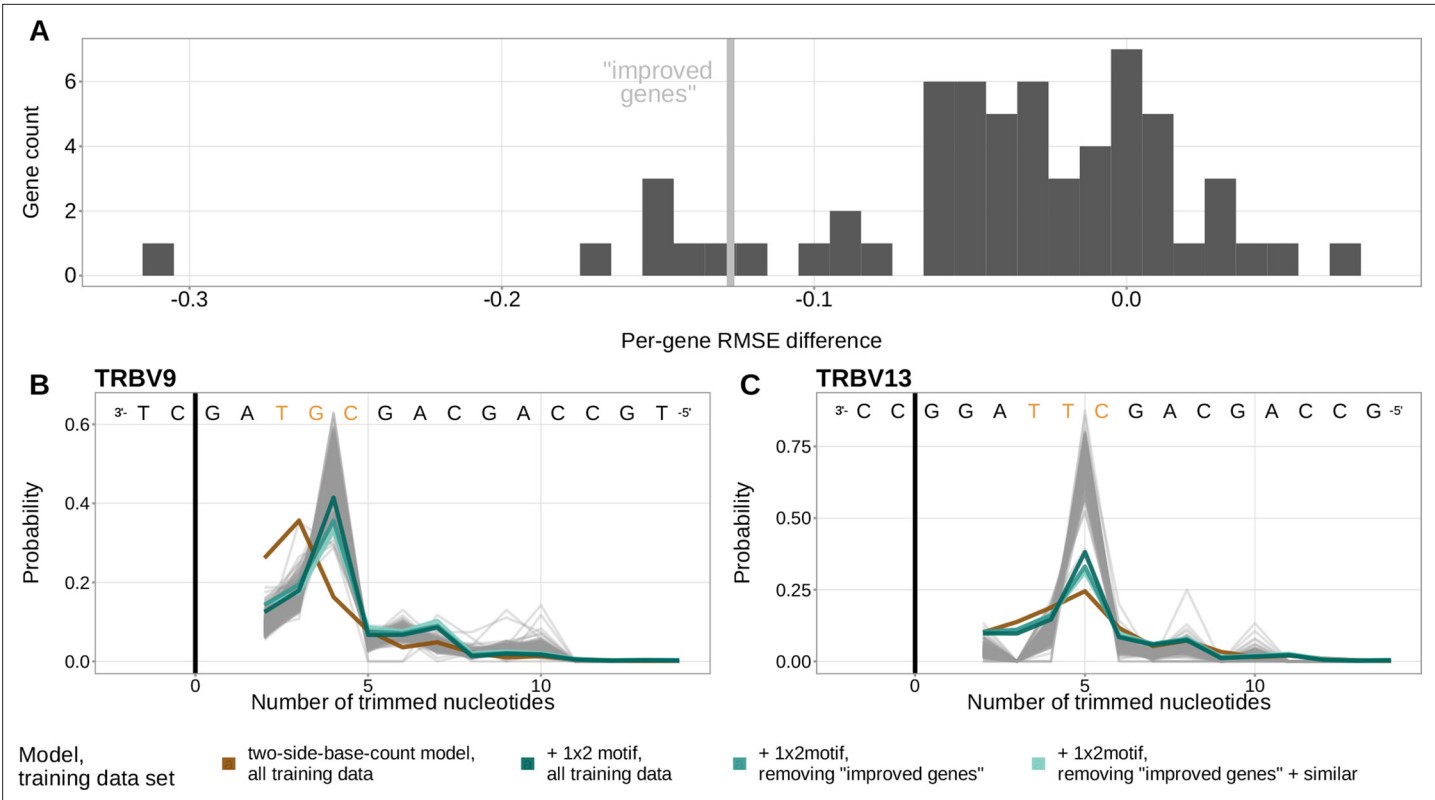

**Figure 5.** The *1×2 motif* coefficients represent a gene-segment-wide trimming motif. (**A**) Distribution of the difference in per-gene root mean squared error (RMSE) between the *1×2 motif + two-side base-count beyond* model and the *two-side base-count* model. V-genes with an RMSE difference less than –0.127 (gray vertical line) were in the lowest 10% of all RMSE differences. These 'improved genes' showed a large RMSE improvement when including motif terms in the model. (**B**) Inferred trimming profiles for *TRBV9*, the gene which showed the largest RMSE improvement in (**A**). *TRBV9* had an RMSE difference of –0.31. (**C**) Inferred trimming profiles for *TRBV13*, the gene which showed the second largest RMSE improvement in (**A**). *TRBV13* had an RMSE difference of –0.15. The inferred trimming profiles for *TRBV9* and *TRBV13* using models which contain motif terms have very low prediction error even when the genes are not included in the model training data set. Gene-specific trimming profiles for each individual in the training data set are shown in gray. The sequence context with the highest probability of trimming (3'-TTC-5' or 3'-TGC-5' from *Figure 4B*) are highlighted in orange.

The online version of this article includes the following source data for figure 5:

**Source data 1.** Per-gene mean squared error difference between the *1×2 motif + two-side base-count beyond* model and the *two-side base-count* model.

**Source data 2.** Inferred and observed trimming profiles for the genes with largest root mean squared error (RMSE) improvement.

whether the inferred coefficients from the V-gene-specific *1×2 motif + two-side base-count beyond* model varied significantly in the context of the non-coding Artemis-locus SNP (rs41298872) that was found to be most strongly associated with increasing the extent of V-gene trimming in our previous work (**Russell et al., 2022b**). As such, we re-defined the model to include an interaction coefficient between the SNP genotype and each model parameter (see Materials and methods). We then used a Bonferroni-corrected significance threshold of 0.0033 (corrected for the total number of interaction coefficients) to evaluate the significance of each interaction coefficient. For each significant interaction coefficient, we concluded that the corresponding model coefficient varied significantly in the context of the SNP genotype.

Using these methods, we found that several of the *1×2 motif + two-side base-count beyond* model coefficients varied significantly in the context of the Artemis-locus SNP rs41298872 (*Figure 6*). Specifically, we found that 3' of the trimming site, the negative effect of A nucleotides on the trimming odds varied in the context of the SNP for the position immediately 3' of the trimming site ($\log_{10}$ interaction coefficient = 0.006, p = 0.0006) and one position away ($\log_{10}$ interaction coefficient = 0.007, p = 0.0006). Further, we found that the negative effect of the count of AT nucleotides 3' of the motif

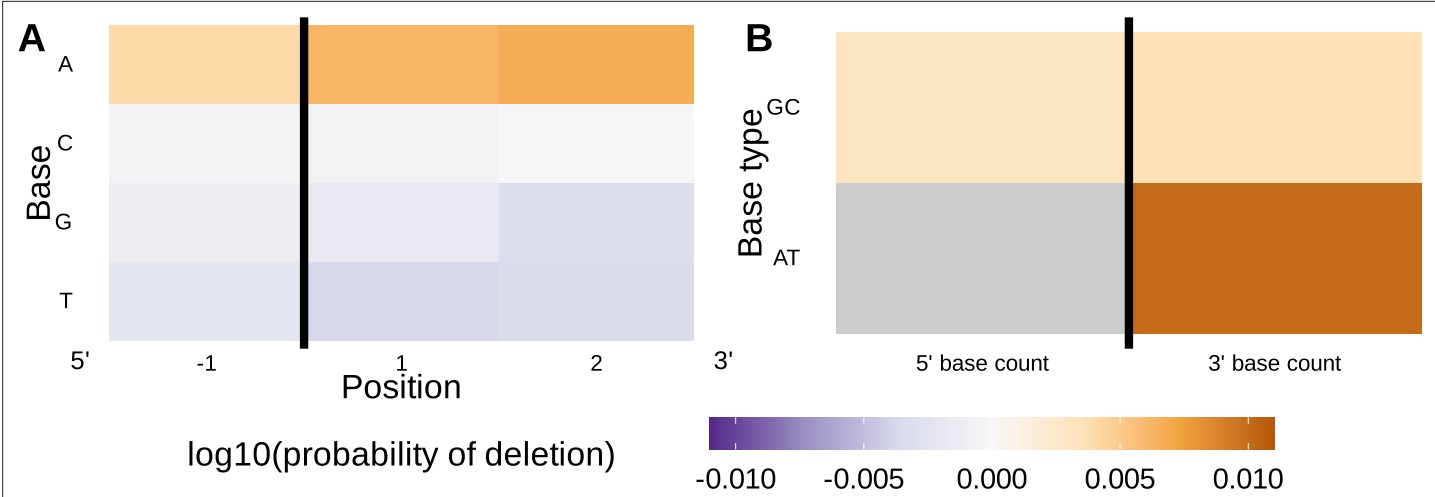

**Figure 6.** Inferred single nucleotide polymorphism (SNP)-parameter-interaction coefficients from fitting the *1×2 motif + two-side base-count beyond* SNP-interaction model. Note that the inferred coefficients for each main parameter (as shown in *Figure 4*) are not displayed here; only the inferred interaction coefficients between the SNP and each parameter are shown. (**A**) Inferred interaction coefficients between rs41298872 SNP genotype and *motif* parameters for 1-nucleotide position 5′ of the trimming site and 2-nucleotide positions 3′ of the trimming site. The interaction coefficients between the SNP genotype and the presence of A nucleotides (at all positions 3′ of the motif) are significant. This figure is a different representation of the information shown in (**A**). (**B**) Inferred interaction coefficients between rs41298872 SNP genotype and *two-side base-count beyond* model coefficients. The interaction coefficients between the SNP genotype and the count of AT nucleotides 3′ of the motif are significant. The interaction coefficient between the SNP genotype and the count of AT nucleotides 5′ of the motif (shown in gray) was not included in this model. The black vertical line corresponds to the trimming site. Each inferred interaction coefficient is given as the change in log10 odds of trimming at a given site resulting from an increase in the feature value and a change in genotype, given that all other features are held constant.

The online version of this article includes the following source data and figure supplement(s) for figure 6:

**Source data 1.** Inferred *1×2 motif + two-side base-count beyond*, single nucleotide polymorphism (SNP) interaction model coefficients.

**Figure supplement 1.** The significance of the 3′-AT-nucleotide count single nucleotide polymorphism (SNP)-interaction coefficient appears to be related to length effects rather than nucleotide content.

varied strongly in the context of the SNP ($\log_{10}$ interaction coefficient = 0.010, p = $1.47 \times 10^{-12}$). No other *motif* or *two-side base-count* coefficients were found to significantly vary.

Because the 3′-side *base-count-beyond* terms parameterize both GC nucleotide content and length in their definition, we were interested in whether the significance of the 3′-AT-nucleotide count SNP variation effect was related to GC nucleotide content, length, or both. To do this, we re-defined the 3′-side *base-count-beyond* parameters to be a proportion instead of raw AT/GC nucleotide counts and included an additional *length* term in the model to remove length-related effects from the inferred 3′-side *base-count-beyond* coefficients. Using this new model, we repeated the analysis and found that the *length* coefficient varied significantly in the context of the SNP ($\log_{10}$ interaction coefficient = 0.005, p = $6.24 \times 10^{-23}$), but the 3′-AT-nucleotide-proportion term did not (*Figure 6—figure supplement 1*). This result is fully consistent with the fact that the Artemis-locus SNP is known to be associated with increasing the extent of trimming (a proxy for length).

### Local sequence context, length, and GC nucleotide content in both directions of the wider sequence can also accurately predict the trimming probabilities of a given sequence from other receptor loci

To validate our previously trained models, we worked with TCRα- and TCRβ-immunosequencing data representing 150 individuals, TCRγ-immunosequencing data representing 23 individuals, and IGH-immunosequencing data representing 9 individuals from three independent validation cohorts. Before analyzing these data, we 'froze' our trained model coefficients in git commit 093610a on our repository. In contrast to the training data cohort, these validation cohorts contain different demographics and were each processed using different sequence annotation methods (see Materials and methods). To explore the potential effects of using a different sequence annotation method, we re-annotated

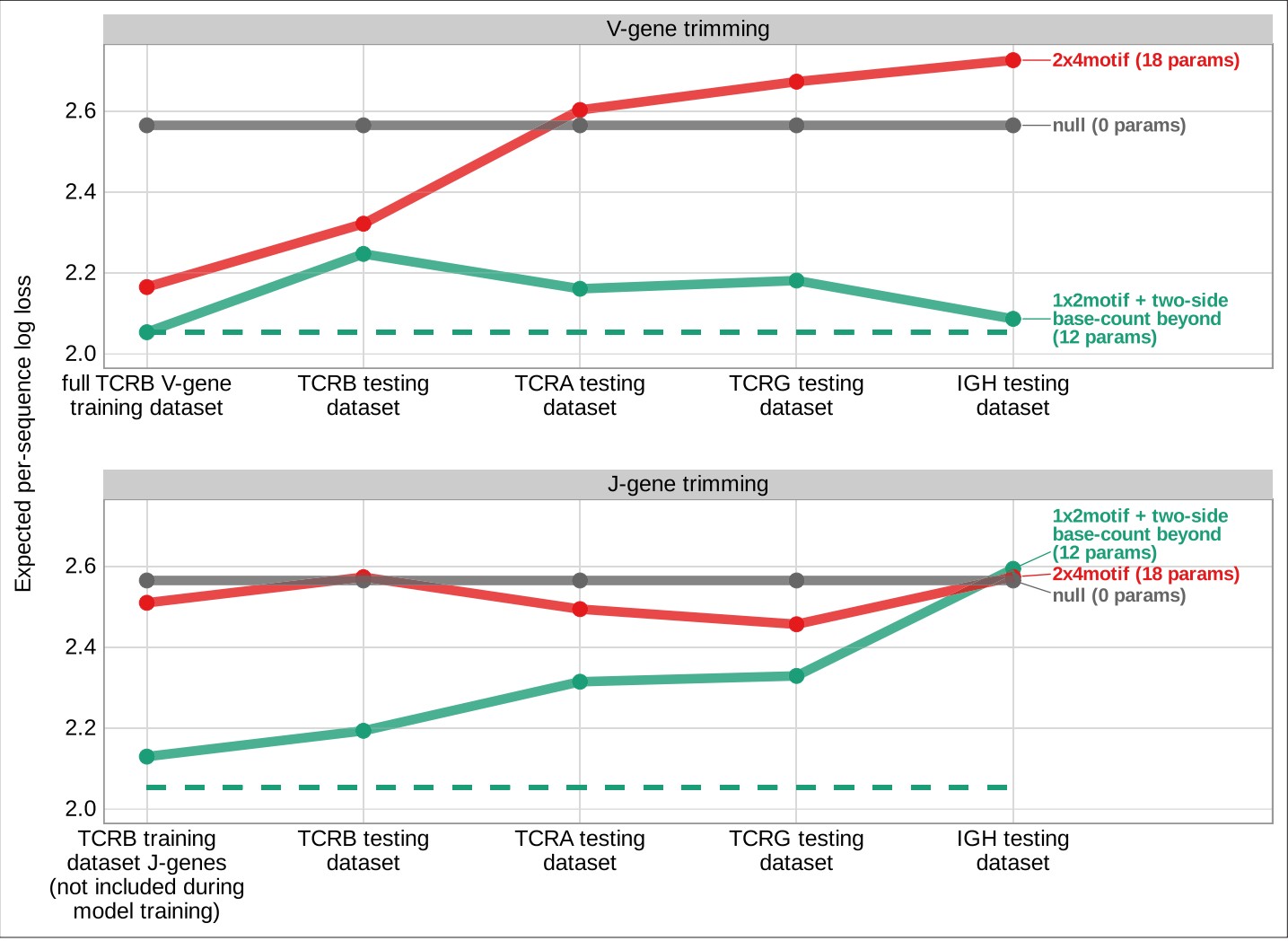

**Figure 7.** Expected per-sequence conditional log loss computed for various models using the T cell receptor β V-gene training data set and non-productive V- and J-gene sequences from several independent testing data sets. Each model was trained using the full non-productive TCRβ V-gene training data set. Lower expected per-sequence log loss corresponds to a better model fit. The *1×2 motif + two-side base-count beyond* model has the best model fit and generalizability across all testing data sets. The horizontal dashed line corresponds to the expected per-sequence log loss of the *1×2 motif + two-side base-count beyond* model computed for V-gene trimming using the non-productive TCRβ V-gene training data set.

The online version of this article includes the following source data and figure supplement(s) for figure 7:

**Source data 1.** Expected per-sequence conditional log loss reported for each model and testing data set.

**Figure supplement 1.** Differing methods of sequence annotation have little to no effect on the model fit or performance.

**Figure supplement 2.** Model performance is similar for productive sequences compared to non-productive sequences from each testing data set.

**Figure supplement 3.** Using the *1×2 motif + two-side base-count beyond* model, the weight of the *two-side base-count beyond* terms are dominant relative to the *1×2 motif* terms for every testing data set.

**Figure supplement 4.** Sequence motifs appear at varying frequencies within the germline *TRB* and *IGH* genes.

the TCRβ training data set using the same annotation method as the TCRα-β testing data and found that it had little to no effect on the model fit or performance (*Figure 7—figure supplement 1*).

To evaluate the performance of the *1×2 motif + two-side base-count beyond* model using these testing data, we used the model coefficients from the previous TCRβ V-gene training run and computed the expected per-sequence conditional log loss of the model using each testing data set (TCRβ V-gene sequences, TCRα V-gene sequences, TCRγ V-gene sequences, IGH V-gene sequences, TCRβ J-gene sequences, etc.). We found that the model has high predictive accuracy (i.e. low expected per-sequence conditional log loss) for both non-productive V- and J-gene sequences from the TCRβ

testing data set (*Figure 7*). The model also extends well to non-productive V- and J-gene sequences from the TCRα and TCRγ testing data sets and to non-productive V-gene sequences from the IGH testing data set. The model has relatively poor predictive accuracy for non-productive IGH J-gene sequences, however. We noted very similar results when validating model performance using productive V- and J-gene sequences from each testing data set (*Figure 7—figure supplement 2*).

We hypothesized that the weight of the *1×2 motif* and *two-side base-count beyond* model terms may vary across each testing data set. To explore this for each data set, we again used the model coefficients from the previous TCRβ V-gene training run and trained a new two-parameter model containing one coefficient scaling the *1×2 motif* terms and a second coefficient scaling the *two-side base-count beyond* terms (see Materials and methods). With this approach, we found that the *two-side base-count beyond* terms were dominant compared to the *1×2 motif* terms for every data set (*Figure 7—figure supplement 3A*). The scale coefficient for the *1×2 motif* terms was very small for several of the data sets, especially the IGH data set, indicating only a weak motif-related signal. The sequence motifs that lead to a large increase in trimming probabilities in the model appear at relatively low frequencies within the germline *IGH* genes (*Figure 7—figure supplement 4*), perhaps explaining the weakness of the motif-related signal. When evaluating the expected per-sequence conditional log loss of these partially re-trained models, we note a small improvement in model fit for each re-trained model compared to the original model (*Figure 7—figure supplement 3B*).

## Discussion

The junctional deletion and insertion steps of the V(D)J recombination process are essential for creating diversity within the TCR repertoire. While the Artemis protein is often regarded as the main nuclease involved in V(D)J recombination, the exact mechanism of nucleotide trimming has yet to be understood in a human system. Using a previously published high-throughput TCRβ sequencing data set, we designed a flexible probabilistic model of nucleotide trimming that allowed us to explore the relative importance of various sequence-level features. While we recognize that these general model features may not capture the full complexity of the trimming mechanism and establish causation, we were primarily interested in identifying mechanistically interpretable features which could confirm and extend our current understanding of the nucleotide trimming process. With this framework, we have (1) revealed a set of sequence-level features which can be used to accurately predict trimming probabilities across various adaptive immune receptor loci, (2) shown that length and GC nucleotide content in both directions of the wider sequence are highly predictive of trimming probabilities, quantifying how double-stranded DNA needs to be able to breathe for trimming to occur, (3) identified a sequence motif that appears to get preferentially trimmed, independent of length- and GC-content-related effects, and (4) demonstrated that a genetic variant within the gene encoding the Artemis protein is associated with changes in several model coefficients.

Specifically, we find that a model containing parameterizations of both local sequence context, length, and GC nucleotide content in both directions of the wider sequence can most accurately predict the trimming probabilities of a given TCRβ gene sequence. In addition to having fewer parameters, this model also had better predictive accuracy than a previously proposed sequence context model (*Murugan et al., 2012*). Models containing other sequence-level parameters such as DNA-shape and length also had relatively worse predictive accuracy. The *TR* and *IG* V(D)J recombination processes, including trimming profiles, have previously been suggested to vary substantially across individuals (*Slabodkin et al., 2021*; *Russell et al., 2022b*). Here, our results support a universal, sequence-based trimming mechanism underlying this variation across *TR* and *IG* loci in humans. Specifically, in addition to TCRβ sequences, we find that local sequence context, length, and GC nucleotide content in both directions of the wider sequence can be used to accurately predict trimming probabilities across TCRα, TCRγ, and IGH sequences. For all of these loci, we find that length and GC nucleotide content are relatively more important than local sequence context terms for making accurate model predictions.

The Artemis protein, in complex with DNA-PKcs, is responsible for opening the DNA hairpin during the early steps of V(D)J recombination to generate a 4-nucleotide-long 3'-single-stranded overhang at the end of each gene, and has been suggested to continue on to trim nucleotides from this resulting DNA structure (*Feeney et al., 1994*; *Nadel and Feeney, 1995*; *Nadel and Feeney, 1997*; *Jackson et al., 2004*; *Gu et al., 2010*; *Chang et al., 2015*; *Chang and Lieber, 2016*; *Zhao et al., 2020*; *Russell*

*et al., 2022b*). The Artemis protein, with and without DNA-PKcs, has been shown to bind single-stranded-to-double-stranded DNA boundaries prior to nicking DNA (*Ma et al., 2002*; *Ma et al., 2005*; *Chang et al., 2015*; *Chang and Lieber, 2016*). While the single-stranded overhang created during hairpin-opening may create a natural single-stranded-to-double-stranded DNA substrate for Artemis binding near the end of the gene sequence, we find that many trimming events occur further into the double-stranded gene sequence. Indeed, previous in vitro DNA nuclease assays involving Artemis have shown that sequence-breathing dynamics are often required to generate a transient single-stranded-to-double-stranded DNA substrate prior to Artemis action (*Chang et al., 2015*). Using our model of nucleotide trimming, we have shown that trimming probabilities are highest for DNA positions closer to the end of the sequence. Because these DNA positions have fewer double-stranded nucleotides on the 3'-side of the trimming site, they may have more capacity for sequence-breathing. On the 5'-side of the trimming site, we find that having a larger number of G-C nucleotides, and perhaps less sequence-breathing capacity, increases the trimming probability. Perhaps this breathing transition can create a transient single-stranded-to-double-stranded DNA substrate that is suitable for Artemis to bind and trim. As such, this finding quantifies sequence-breathing effects that were previously identified through in vitro DNA nuclease assay studies involving Artemis (*Chang et al., 2015*).

Independent of GC-content-related effects, we have also identified a gene-segment-wide sequence motif that appears to get preferentially trimmed. This motif is suggestive of sequence-specific nucleolytic activity, however, Artemis is widely regarded as a structure-specific nuclease as opposed to a nuclease that binds specific DNA sequences (*Ma et al., 2005*; *Chang et al., 2015*; *Chang and Lieber, 2016*; *Yosaatmadja et al., 2021*). This suggests that either (1) Artemis actually does possess some ability to recognize specific nucleotides, (2) the observed sequence motif is serving as a proxy for DNA structure induced by the motif, or (3) another nuclease, in addition to Artemis, is responsible for the sequence-specific trimming we observe. However, because the strength of this sequence motif signal varied across receptor loci, further work will be required to explore its mechanistic basis and presence.

We found that several model coefficients related to local sequence context, length, and GC nucleotide content in both directions of the wider sequence varied significantly in the context of the noncoding Artemis-locus SNP rs41298872. We previously identified this Artemis-locus SNP as being associated with increasing the extent of TCRβ V- and J-gene trimming (*Russell et al., 2022b*). While many previous studies have reported a high consistency of TCRβ trimming profiles across individuals (*Murugan et al., 2012*; *Marcou et al., 2018*; *Sethna et al., 2020*), our results begin to explore how the trimming mechanism may vary across individuals in the context of Artemis genetic variation. We reported that trimming probabilities decrease as the number of double-stranded nucleotides 3' of the trimming site increases. In the context of the SNP rs41298872, we found that as the number of double-stranded AT nucleotides 3' of the trimming site increases, the trimming probabilities do not decrease as quickly. This suggests that individuals homozygous (or heterozygous) for rs41298872 may be more capable of trimming at positions that have a larger number of double-stranded nucleotides 3' of the trimming site, especially if the additional nucleotides are AT bases. This may be possible if, for example, rs41298872 increases Artemis expression. If there is more Artemis available, then trimming at less optimal positions (i.e. positions further into the sequence which have less breathing) may be possible. Additional work will be required to define the relationship between rs41298872 genotype and Artemis expression.

We also identified several local sequence context coefficients that varied in the context of rs41298872, however, their mechanistic interpretation remains unclear. Earlier, we noted that A nucleotides 3' of the trimming site have a negative effect on the trimming probability while T nucleotides have a strong positive effect. In the context of rs41298872, we found that the magnitude of the negative effect of 3' A nucleotides on the trimming probability was reduced. This may suggest that individuals homozygous (or heterozygous) for rs41298872 may trim in a less motif-dependent fashion, and are instead more reliant on sequence openness 3' of the trimming site. In this way, having A or T nucleotides 3' of the trimming site would create a more open local sequence for trimming.

There are several key limitations of our approach which are intrinsic to the use of adaptive immune receptor repertoire data. First, we have used trimming statistics from non-productive rearrangements as a means of studying the nucleotide trimming process in the absence of selection. Non-productive sequences can be sequenced as part of the repertoire when they are present within a cell expressing a productive rearrangement that survived the selection process. While we are not aware of a mechanism

through which non-productive and productive rearrangements within a single cell could be correlated, we also acknowledge that the repertoire of non-productive rearrangements may be an imperfect proxy for a pre-selection repertoire. However, as is common in the literature (*Robins et al., 2010*; *Murugan et al., 2012*; *Marcou et al., 2018*; *Sethna et al., 2019*; *Sethna et al., 2020*), we assume that the two recombination events are independent and that the non-productive rearrangements reflect the statistics of the repertoire prior to selection. Next, because many V(D)J rearrangement scenarios can give rise to the same final nucleotide sequence, possible error related to the annotation of each sequence may have restricted our ability to model the actual trimming distributions of each gene. Although we cannot rule out some effect of incorrect sequence annotation on our model inferences, we found that the exact sequence annotation method used, including sampling from the posterior distribution of rearrangement events, had little to no effect on the model fit or performance.

In summary, we have found that local sequence context, length, and the GC nucleotide content in both directions of the wider sequence can accurately predict the trimming probabilities of *TR* and *IG* gene sequences. These results refine our understanding of how nucleotides are trimmed during V(D)J recombination. The sequence-level features identified here lay the groundwork for further exploration into the trimming mechanism and how it may vary across individuals. Such insights will provide another step toward understanding how V(D)J recombination generates diverse receptors and supports a powerful, unique immune response in humans.

## Materials and methods
### Training data set
TCRβ repertoire sequence data for 666 healthy bone marrow donor subjects was downloaded from the Adaptive Biotechnologies immuneACCESS database using the link provided in the original publication (*Emerson et al., 2017*). V(D)J recombination scenarios were assigned to each sequence for each individual using the IGoR software (version 1.4.0) (*Marcou et al., 2018*) as follows. The IGoR software can learn unbiased V(D)J recombination statistics from immune sequence reads. Using these statistics, IGoR can output a list of potential recombination scenarios with their corresponding likelihoods for each sequence. As such, using the default IGoR V(D)J recombination statistics, the 10 highest probability V(D)J recombination scenarios were inferred for each TCRβ-chain sequence in the training data set (*Marcou et al., 2018*). We then annotated each TCRβ-chain sequence with a single V(D)J recombination scenario by sampling from these 10 scenarios according to the posterior probability of each scenario. We filtered these sequences for rearrangements which contained more than 1 trimmed nucleotide and less than 15 trimmed nucleotides (see the 'Notation' section for further details). We further subset the data to include only non-productive sequences, and used these data for all subsequent model training. After these processing and filtering steps, we used V-gene trimming length distributions from 21,193,153 non-productive sequences for all model training. To test each trained model, we used V-gene trimming length distributions from the remaining 107,121,841 productive sequences (as described in Appendix 3). From this same data set, we also used J-gene trimming length distributions from 107,255,406 productive sequences and 20,204,801 non-productive sequences to test each model.

### Testing data sets
#### TCRα and TCRβ testing data sets
Annotated TCRα and TCRβ repertoire sequence data for 150 healthy subjects was downloaded using the link provided in the original publication (*Russell et al., 2022b*). In contrast to the training data cohort, this cohort contains different demographics, shallower RNA-seq-based TCR sequencing, and was processed using a different sequence annotation methods (i.e. TCRdist [version 0.0.2] [*Dash et al., 2017*] as described in a previous publication [*Russell et al., 2022b*]). Sequences were split into non-productive and productive groups for model validation. From the TCRα data set, we used V-gene trimming length distributions from 123,496 non-productive sequences and 862,096 productive sequences and J-gene trimming length distributions from 141,451 non-productive sequences and 1,101,114 productive sequences to test each model. From the TCRβ data set, we used V-gene trimming length distributions from 64,738 non-productive sequences and 1,435,153 productive

sequences and J-gene trimming length distributions from 59,608 non-productive sequences and 1,496,953 productive sequences to test each model.

### TCRγ testing data set

Annotated TCRγ repertoire sequence data for 23 healthy bone marrow donor subjects was downloaded from the Adaptive Biotechnologies immuneACCESS database (*Robins and Pearson, 2015*). Sequences were split into non-productive and productive groups for model validation. We used V-gene trimming length distributions from 2,403,293 non-productive sequences and 1,002,662 productive sequences and J-gene trimming length distributions from 568,824 non-productive sequences and 250,493 productive sequences to test each model.

### IGH testing data sets

Annotated IgG class non-productive IGH repertoire sequence data for nine healthy subjects was obtained from the authors of a previous publication (*Spisak et al., 2020*). The raw sequence data is available using the link provided in the original publication (*Briney et al., 2019*). In contrast to the training data cohort, this cohort contains different demographics, shallower RNA-seq based IGH-sequencing, and was processed using a different sequence annotation method (i.e. a combination of Immcantation [*Vander Heiden et al., 2014*] and IgBlast [*Ye et al., 2013*] as described in a previous publication [*Spisak et al., 2020*]). Further, these data are restricted to rearrangements that lead to a clonal family with at least six members.

Likewise, productive IGH repertoire sequence data for four healthy subjects was downloaded using the link provided in the original publication (*Jaffe et al., 2022*) and the sequences were annotated using partis (version 0.16.0) (*Ralph and Matsen, 2016*). Due to the large size of this data set, 100k sequences were randomly sampled from the original data set prior to model validation. For both IGH data sets, only a single sequence from each inferred clonal family was included in each model testing data set. From these data sets, we used V-gene trimming length distributions from 160,714 non-productive sequences and 32,245 productive sequences and J-gene trimming length distributions from 297,298 non-productive sequences and 74,884 productive sequences to test each model.

## Artemis-locus SNP data set

Genome-wide SNP array data corresponding to 611 of the training data set individuals was downloaded from The database of Genotypes and Phenotypes (accession number: phs001918). Details of the SNP array data set, genotype imputation, and quality control have been described previously (*Martin et al., 2020*). We only used SNP data corresponding to the Artemis locus (rs41298872) which we previously found to be strongly associated with increasing the extent of V-gene trimming (*Russell et al., 2022b*).

## Notation

Let $I$ be a set of individuals. For each subject $i \in I$, assume we have a TCR repertoire consisting of sequences indexed by $k$ such that $k = 1, \ldots, K_i$. We assume that each sequence can be unambiguously annotated with being from a specific V-gene and J-gene sequence, and having a number of deleted nucleotides from each gene. For modeling purposes, we combine *TRB* V-gene or J-gene alleles that have identical terminal nucleotide sequences (last 24 nucleotides of each sequence) into *TRB* V-gene allele groups and *TRB* J-gene allele groups. As such, each TCR sequence is annotated with being from a V-gene allele group and J-gene allele group. Because we are requiring that each gene allele group originates from the same *TRB* V-gene or J-gene, there may still be overlap in terms of sequence identity between allele groups. For simplicity, we orient all sequences in the 5'-to-3' direction, and use the top strand for V-gene sequences and the bottom strand for J-gene sequences. We will be introducing modeling methods as they relate to V-genes and V-gene trimming, however, with this sequence orientation, the same methods can be applied to J-genes and J-gene trimming. We will use $\sigma$ to represent a gene sequence oriented in the 5'-to-3' direction and $n$ to represent the number of nucleotides deleted from the 3' end of this sequence as we describe our modeling.

We are interested in modeling the probability of trimming $n$ nucleotides from a given gene sequence $\sigma$, $P(n \mid \sigma)$. We can define an empirical conditional probability density function to estimate this probability. To start, we can uniformly sample from any given individual's repertoire. Let $S$ be a

random variable that represents the gene-allele-group sequence from such a sample. Let $N$ be a random variable that represents the number of deleted nucleotides, which for notational convenience we assume take on a non-negative integer value (nonsensical values will have probability zero). Let $0 \leq C^{(i)}(\sigma) \leq K_i$ represent the number of TCRs that use gene allele group $\sigma$. Let $0 \leq C^{(i)}(n, \sigma) \leq K_i$ represent the number of TCRs that have gene allele group $\sigma$ and $n$ gene nucleotides deleted. With these data, we can form the empirical conditional probability density function:

$$P_{\text{emp}}(N = n \mid S = \sigma, i) = \frac{C^{(i)}(n, \sigma)}{C^{(i)}(\sigma)}. \tag{1}$$

Using these TCRβ repertoire data, we want to model the influence of various sequence-level parameters on $P(n \mid \sigma)$. With this assumption, let $L$ and $U$ be lower and upper bounds, respectively, on $n$ such that $N' = \{L, \ldots, U\}$ is the set of all reasonable nucleotide deletion amounts. The precise location of hairpin opening and its relationship to deletion is unclear. Hence, we have chosen to define $L = 2$ since smaller trimming amounts may result from an alternative, hairpin-opening-position-related (or other) trimming mechanism. Likewise, we have chosen to define $U = 14$ since trimming amounts greater than 14 nucleotides are uncommon and could also result from an alternative trimming mechanism. We will subset the training data set, after IGoR annotation (see details in a previous section), such that we will only consider TCRs that have $2 \leq n \leq 14$. Similarly, the one existing analysis (*Murugan et al., 2012*) exploring the relationship between sequence context and nucleotide trimming only considered TCRs that had $2 \leq n \leq 12$ for their modeling. We summarize all of the notation discussed in this section, as well as in the following sections, in *Appendix 1—table 1*.

## V(D)J recombination modeling assumptions

For our model, we make the following assumptions about V(D)J recombination biology:

1. During the V(D)J recombination process, the gene DNA hairpin is nicked open by a single-stranded break (*Gauss and Lieber, 1996*; *Nadel and Feeney, 1997*; *Ma et al., 2002*; *Jackson et al., 2004*; *Lu et al., 2007*).
2. This hairpin nick occurs at the +2 position, leading to a 4-nucleotide-long 3′-single-stranded-overhang (the 2 nucleotides furthest 3′ are considered P-nucleotides) (*Ma et al., 2002*; *Lu et al., 2007*). We will discuss a sensitivity analysis to this assumption, which showed that the assumed hairpin-nick position had little impact on our model fitting, in the appendix.
3. If any nonzero amount of the original gene sequence is deleted, all P-nucleotides will also be deleted (*Gauss and Lieber, 1996*; *Srivastava and Robins, 2012*).
4. Nucleotide trimming occurs before N-insertion.

With these assumptions, we can resolve the nucleotide sequence on both sides of the trimming site and define mechanistically interpretable model features using these two sequences. Specifically,

**Table 1.** Summary of all parameter-specific coefficients and covariate functions for a trimming site $n$ and gene sequence $\sigma$.

Here, $a$ and $b$ represent the number of nucleotides 5′ and 3′ of the trimming site to be included in the 'trimming motif,' respectively, and $c$ represents the number of nucleotides 5′ of the trimming site to be included in the base-count.

| Parameter | Model coefficient variables | Parameter-specific covariate function |
|---|---|---|
| *Motif* parameters | $\beta^{\texttt{motif}}$ coefficients | $f_1(n, \sigma; \beta^{\texttt{motif}}, a, b)$ *(Equation 14)* |
| *Base-count-beyond* parameters | $\beta^{\texttt{AT}}$ and $\beta^{\texttt{GC}}$ coefficients | $f_2(n, \sigma; \beta^{\texttt{AT}}, \beta^{\texttt{GC}}, a, b, c)$ *(Equation 17)* |
| *DNA-shape* parameters | $\beta^{\texttt{shape}}$ coefficients | $f_3(n, \sigma; \beta^{\texttt{shape}}, a, b)$ *(Equation 19)* |
| *Length* parameters | $\beta^{\texttt{ldist}}$ coefficients | $f_4(n, \sigma; \beta^{\texttt{ldist}})$ *(Equation 20)* |

we define a 'trimming motif' consisting of several nucleotides on either side of the trimming site, the predicted 'DNA-shape' of the nucleotides and bonds in close proximity to the trimming site, the counts of GC or AT nucleotides on either side of the trimming site beyond the 'trimming motif' region (e.g. the 'two-side base-count beyond'), and the sequence-independent 'length' from the end of the gene to the trimming site (see Appendix 2 for further details). An example of how an arbitrary V-gene sequence is transformed into features for modeling is shown in *Figure 1*. We will assume that observations can be drawn from a model in which these features vary across trimming lengths $n$ for a given gene allele group $\sigma$. We can then explore the influence of these features on the probability of trimming at a certain site given a gene sequence.

## Defining a model covariate function

With the features summarized above, we can define a model covariate function $f$ than contains any unique combination of parameter-specific covariate functions (*Table 1*). This function $f$ will be the sum of each of the desired parameter-specific covariate functions. This framework allows us to generalize the existing PWM model (*Murugan et al., 2012*) to a model that allows for arbitrary sequence features. For example, we replicate this PWM model using the model covariate function, $f_1(n, \sigma; \beta^{\texttt{motif}}, a = 2, b = 4)$, where $n$ represents the number of trimmed nucleotides, $\sigma$ represents the gene-allele-group sequence, $\beta^{\texttt{motif}}$ represents *motif*-specific parameter coefficients, and $a$ and $b$ are non-negative integer values that represent the number of nucleotides 5' and 3' of the trimming site, respectively, that are included in the 'trimming motif'. This function is described further in (*Equation 14*). To extend this model to a model containing *motif* parameters and *base-count-beyond* parameters, the model covariate function will be

$$f(n, \sigma; \beta^{\texttt{motif}}, \beta^{\texttt{AT}}, \beta^{\texttt{GC}}, a, b, c) := f_1(n, \sigma; \beta^{\texttt{motif}}, a, b) + f_2(n, \sigma; \beta^{\texttt{AT}}, \beta^{\texttt{GC}}, a, b, c) \qquad (2)$$

where $f_2$ represents the *base-count-beyond* model covariate function (*Equation 17*), $\beta^{\texttt{AT}}$ and $\beta^{\texttt{GC}}$ represent *base-count-beyond*-specific parameter coefficients, and $c$ represents the number of nucleotides 5' of the trimming site to be included in the base-count. We will use this *motif* and *base-count-beyond* model example to discuss the model formulation in the following sections, however, many other parameter combinations are possible. We will not define a model covariate function that contains two parameters that model the same feature. For example, *length* and *base-count-beyond* coefficients will never be included in a model covariate function together (since they both parameterize length). Likewise, *motif* and *DNA-shape* coefficients will never both be included in a model covariate function.

## Predicting trimming probabilities using conditional logistic regression

We will be using the *motif* and *base-count-beyond* parameters given by (*Equation 2*) as examples for the remainder of this section, however, we could also formulate a model with any other parameter of interest, as described in the previous section (*Table 1*). As such, we can fit a conditional logit model which posits that

$$P(n \mid \sigma; \beta^{\texttt{motif}}, \beta^{\texttt{AT}}, \beta^{\texttt{GC}}, a, b, c) := \frac{\exp(f(n, \sigma; \beta^{\texttt{motif}}, \beta^{\texttt{AT}}, \beta^{\texttt{GC}}, a, b, c))}{\sum_{n' \in N'} \exp(f(n', \sigma; \beta^{\texttt{motif}}, \beta^{\texttt{AT}}, \beta^{\texttt{GC}}, a, b, c))} \qquad (3)$$

where $N'$ is the set of all reasonable trimming lengths, $a$ and $b$ represent the number of nucleotides 5' and 3' of the trimming site to be included in the 'trimming motif,' respectively, $c$ represents the number of nucleotides 5' of the trimming site to be included in the base-count parameters, and $f(n, \sigma; \beta^{\texttt{motif}}, \beta^{\texttt{AT}}, \beta^{\texttt{GC}}, a, b, c)$ is the model covariate function for the *motif* and *base-count-beyond* model given by (*Equation 2*). We will let $P(n \mid \sigma; \beta^{\texttt{motif}}, \beta^{\texttt{AT}}, \beta^{\texttt{GC}}, a, b, c)$ denote the conditional probability that a given gene will be trimmed by $n$ nucleotides.

Let $y_{ik\sigma n}$ equal 1 if a gene allele group $\sigma$ is trimmed by $n$ nucleotides for TCR $k$ from subject $i$, and equal 0 otherwise. With this, we can define a likelihood function, $L(\beta^{\texttt{motif}}, \beta^{\texttt{AT}}, \beta^{\texttt{GC}}, a, b, c)$, such that for a random sample of subjects, $L(\beta^{\texttt{motif}}, \beta^{\texttt{AT}}, \beta^{\texttt{GC}}, a, b, c)$, is the likelihood of the model parameters, $\beta^{\texttt{motif}}$, $\beta^{\texttt{AT}}$, and $\beta^{\texttt{GC}}$, given that we observed a set of trimming amounts for a set of given genes. As such, the log-likelihood function can be written as

$$\log L(\beta^{\texttt{motif}}, \beta^{\text{AT}}, \beta^{\text{GC}}, a, b, c) = \sum_{i,k,\sigma,n} y_{ik\sigma n} \cdot \log P(n \mid \sigma; \beta^{\texttt{motif}}, \beta^{\text{AT}}, \beta^{\text{GC}}, a, b, c)$$

where $P(n \mid \sigma; \beta^{\texttt{motif}}, \beta^{\text{AT}}, \beta^{\text{GC}}, a, b, c)$ is given by (*Equation 3*). Instead of maximizing this log-likelihood directly, we may wish to aggregate the data to reduce the number of observations and simplify model fitting. Recall that for subject $i$, $C^{(i)}(\sigma)$ represents the number of TCRs which use gene allele group $\sigma$ and $C^{(i)}(n, \sigma)$ represents the number of TCRs which have gene allele group $\sigma$ and $n$ gene nucleotides deleted. As such, $C^{(i)}(n, \sigma)$ is the count of observations which will have the same trimming probabilities $P(n \mid \sigma; \beta^{\texttt{motif}}, \beta^{\text{AT}}, \beta^{\text{GC}}, a, b, c)$ and will have been trimmed by $n$ for subject $i$ and gene allele group $\sigma$. Thus, using this aggregated data from all subjects $i \in I$, we can re-write the log-likelihood function equivalently as

$$\log L(\beta^{\texttt{motif}}, \beta^{\text{AT}}, \beta^{\text{GC}}, a, b, c) = \sum_{i,\sigma,n} C^{(i)}(n, \sigma) \cdot \log P(n \mid \sigma; \beta^{\texttt{motif}}, \beta^{\text{AT}}, \beta^{\text{GC}}, a, b, c). \tag{4}$$

As above, for a random sample of subjects, $L(\beta^{\texttt{motif}}, \beta^{\text{AT}}, \beta^{\text{GC}}, a, b, c)$ is the likelihood of the model parameters, $\beta^{\texttt{motif}}$, $\beta^{\text{AT}}$, and $\beta^{\text{GC}}$, given that we observed a set of trimming amounts for a set of given genes.

With this likelihood formulation, all observations in the sample get uniform treatment in the construction of the likelihood. However, subjects may differ in their repertoire size and composition for reasons other than trimming. For example, it is known that gene usage differs across subjects. Thus, to avoid having these differences pollute our $\beta^{\hat{\texttt{motif}}}$, $\beta^{\hat{\text{AT}}}$, and $\beta^{\hat{\text{GC}}}$ inference, we propose a subject and gene weighting scheme.

As such, we can define the expected likelihood of a process where we first draw a subject $i$ uniformly at random, then we sample TCR sequences from their repertoire according to a given distribution, as follows. For a single TCR sequence from such a sample, let $S$ be a random variable representing the gene of the sequence, and let $N$ be a random variable representing the number of deleted nucleotides. We can sample each TCR sequence with probability $P_{\text{samp}}(N = n, S = \sigma)$ which we will specify later. Also, given random $S$ and $N$, the log-likelihood of the model parameters, $\beta^{\texttt{motif}}$, $\beta^{\text{AT}}$, and $\beta^{\text{GC}}$, is given by

$$\log L(\beta^{\texttt{motif}}, \beta^{\text{AT}}, \beta^{\text{GC}}, a, b, c; N, S) = \log P(N \mid S; \beta^{\texttt{motif}}, \beta^{\text{AT}}, \beta^{\text{GC}}, a, b, c).$$

With this, the expected log-likelihood of the model parameters, $\beta^{\texttt{motif}}$, $\beta^{\text{AT}}$, and $\beta^{\text{GC}}$ given this random sample is given by

$$
\begin{aligned}
E[\log &L(\beta^{\texttt{motif}}, \beta^{\text{AT}}, \beta^{\text{GC}}, a, b, c) \mid I = i] \\
&= \sum_{n,\sigma} P_{\text{samp}}(N = n, S = \sigma) \\
&\qquad\qquad \cdot \log P(N = n, S = \sigma; \beta^{\texttt{motif}}, \beta^{\text{AT}}, \beta^{\text{GC}}, a, b, c) \\
&= \sum_{n,\sigma} P_{\text{samp}}(N = n, S = \sigma) \\
&\qquad\qquad \cdot \log P(N = n \mid S = \sigma; \beta^{\texttt{motif}}, \beta^{\text{AT}}, \beta^{\text{GC}}, a, b, c).
\end{aligned}
$$

We can define a new, weighted log-likelihood function, $\log L_{\text{expected}}(\beta^{\texttt{motif}}, \beta^{\text{AT}}, \beta^{\text{GC}}, a, b, c)$, equivalent to this expected log-likelihood:

$$\log L_{\text{expected}}(\beta^{\texttt{motif}}, \beta^{\text{AT}}, \beta^{\text{GC}}, a, b, c) := \sum_{i} E[\log L(\beta^{\texttt{motif}}, \beta^{\text{AT}}, \beta^{\text{GC}}, a, b, c) \mid I = i]. \tag{5}$$

For a random sample of subjects, the weighted likelihood, $L_{\text{expected}}(\beta^{\texttt{motif}}, \beta^{\text{AT}}, \beta^{\text{GC}}, a, b, c)$, represents the likelihood of the model parameters, $\beta^{\texttt{motif}}$, $\beta^{\text{AT}}$, and $\beta^{\text{GC}}$, given that we observed a set of trimming amounts for a given set of gene allele groups after weighting observations according to the sampling procedure $P_{\text{samp}}(N = n, S = \sigma)$. We can use whichever sampling procedure, $P_{\text{samp}}(N = n, S = \sigma)$, we

want. For example, recall that we originally formed the empirical conditional PDFs in (*Equation 1*) for each subject $i$ by uniformly sampling from each TCR repertoire to get a total repertoire size of $K_i$:

$$P_{\text{emp}}(N = n \mid S = \sigma, i) = \frac{C^{(i)}(n, \sigma)}{C^{(i)}(\sigma)},$$

$$P_{\text{emp}}(S = \sigma \mid i) = \frac{C^{(i)}(\sigma)}{K_i},$$

and

$$P_{\text{emp}}(i) = \frac{1}{I}.$$

With this, we can define a sampling procedure equivalent to this empirical joint PDF as follows:

$$
\begin{aligned}
P_{\text{samp}}(N = n, S = \sigma) \quad &:= P_{\text{emp}}(N = n, S = \sigma) \\
&= P_{\text{emp}}(n \mid \sigma, i) \cdot P_{\text{emp}}(\sigma \mid i) \cdot P_{\text{emp}}(i)
\end{aligned}
\tag{6}
$$

With this sampling procedure,

$$
\begin{aligned}
\log L_{\text{expected}}(\beta^{\texttt{motif}}, \beta^{\texttt{AT}}, \beta^{\texttt{GC}}, a, b, c) \\
= \sum_{i,\sigma,n} P_{\text{emp}}(n \mid \sigma, i) \cdot P_{\text{emp}}(\sigma \mid i) \cdot P_{\text{emp}}(i) \\
\cdot \log P(n \mid \sigma; \beta^{\texttt{motif}}, \beta^{\texttt{AT}}, \beta^{\texttt{GC}}, a, b, c).
\end{aligned}
\tag{7}
$$

As such, each subject, instead of each observation, gets uniform treatment in the construction of the weighted likelihood.

While this procedure would correct for individual subjects having different repertoire sizes, it does not account for gene usage differences. To avoid having these differences pollute our $\beta^{\hat{\texttt{motif}}}$, $\beta^{\hat{\texttt{AT}}}$, and $\beta^{\hat{\texttt{GC}}}$ inference, we propose a subject-independent gene-allele-group sampling scheme. While we could use any distribution on $\sigma$, including a uniform weight by gene allele groups, we have chosen to define:

$$P_{\text{marg}}(\sigma) = \frac{1}{I} \sum_i P_{\text{emp}}(\sigma \mid i).$$

We can reformulate the sampling procedure which is an empirical average per-gene-allele-group frequency such that:

$$P_{\text{samp}}(N = n, S = \sigma) := P_{\text{emp}}(n \mid \sigma, i) \cdot P_{\text{marg}}(\sigma) \cdot P_{\text{emp}}(i). \tag{8}$$

With this subject-independent gene sampling procedure, we can define a weighted likelihood $L_W(\beta^{\texttt{motif}}, \beta^{\texttt{AT}}, \beta^{\texttt{GC}}, a, b, c)$ such that

$$
\begin{aligned}
\log L_W(\beta^{\texttt{motif}}, \beta^{\texttt{AT}}, \beta^{\texttt{GC}}, a, b, c) \\
:= \sum_{i,\sigma,n} P_{\text{emp}}(n \mid \sigma, i) \cdot P_{\text{marg}}(\sigma) \cdot P_{\text{emp}}(i) \\
\cdot \log P(n \mid \sigma; \beta^{\texttt{motif}}, \beta^{\texttt{AT}}, \beta^{\texttt{GC}}, a, b, c)
\end{aligned}
\tag{9}
$$

As such, each gene and each subject get uniform treatment in the construction of the weighted likelihood.

From here, we can maximize this weighted log-likelihood, $\log L_W(\beta^{\texttt{motif}}, \beta^{\texttt{AT}}, \beta^{\texttt{GC}}, a, b, c)$, to estimate the log-probabilities $\beta^{\texttt{motif}}$, $\beta^{\texttt{AT}}$, and $\beta^{\texttt{GC}}$, where $\beta^{\texttt{motif}}$ is equivalent to a (log) position-weight-matrix. To estimate each coefficient, we can solve the weighted maximum likelihood estimation problem:

$$(\beta^{\hat{\texttt{motif}}}, \beta^{\hat{\texttt{AT}}}, \beta^{\hat{\texttt{GC}}}) = \text{argmax}_{\beta^{\texttt{motif}}, \beta^{\texttt{AT}}, \beta^{\texttt{GC}}} \log L_W(\beta^{\texttt{motif}}, \beta^{\texttt{AT}}, \beta^{\texttt{GC}}, a, b, c) \tag{10}$$

using the `mclogit` package in R. We can formulate a weighted maximum likelihood problem in a similar way for any model covariate function $f$ containing a unique combination of parameter-specific covariate functions (*Table 1*).

We compare our inferred coefficients to the existing PWM model which was designed and trained using least squares (*Murugan et al., 2012*). When replicating this model using our methods described above (i.e. the *2×4 motif* model), we note highly similar results (*Figure 2—figure supplement 1*).

## Evaluating model fit and generalizability across genes

In order to evaluate the model fit and generalizability of each model, we use a variety of training and testing data sets to train each model and calculate the log loss. We will describe our general model evaluation procedure here. We describe variations of this general model evaluation procedure in Appendix 3. Let $\mathbf{T}$ represent a training data set and $\mathbf{H}$ represent a held-out testing data set. With the training set $\mathbf{T}$, we can train each model of interest as described above in (*Equation 10*). After this model fitting, we can calculate the expected per-sequence conditional log loss of the model with given coefficients, $\mathcal{M}$, for a given held-out testing set, $\mathbf{H}$, such that

$$\ell(\mathcal{M} \mid \mathbf{H})$$
$$:= -\sum_{i,\sigma,n} P_{\mathrm{emp}_{\mathbf{H}}}(n,\sigma,i) \cdot \log P(n \mid \sigma; \mathcal{M})$$
$$= -\sum_{i,\sigma,n} P_{\mathrm{emp}_{\mathbf{H}}}(n \mid \sigma, i) \cdot P_{\mathrm{emp}_{\mathbf{H}}}(\sigma \mid i) \cdot P_{\mathrm{emp}_{\mathbf{H}}}(i)$$
$$\cdot \log P(n \mid \sigma; \mathcal{M})$$

(11)

where $i$ represents a subject, $n$ represents a trimming length, and $\sigma$ represents a gene allele group. Because we are incorporating the empirically observed frequency of each subject, trimming length, and gene allele group within each 'held-out testing set,' $P_{\mathrm{emp}_{\mathbf{H}}}(n,\sigma,i)$, in this formulation, the expected per-sequence conditional log loss values are guaranteed to be directly comparable between held-out testing sets with varying compositions. Models that have lower expected per-sequence conditional log loss will indicate that the model has a better fit.

## Assessing significance of model coefficients

During model fitting, we estimated the model coefficients $\beta^{\hat{\mathrm{motif}}}$, $\beta^{\hat{\mathrm{AT}}}$, and $\beta^{\hat{\mathrm{GC}}}$ by maximizing the weighted likelihood function given by (*Equation 9*). To measure the significance of each of these model coefficients $\hat{\beta} \in \{\beta^{\hat{\mathrm{motif}}}, \beta^{\hat{\mathrm{AT}}}, \beta^{\hat{\mathrm{GC}}}\}$ we want to test whether each coefficient $\hat{\beta} = 0$. To do this, we can first estimate the standard error of each inferred coefficient using a clustered bootstrap (with subject-gene pairs as the sampling unit). As such, for each bootstrap iterate, we sampled subject-gene pairs from the full V-gene training data set with replacement. Using this re-sampled data, we maximized the weighted likelihood function given by (*Equation 9*) to re-estimate each coefficient. We repeated this bootstrap process 1000 times and used the resulting 1000 coefficient estimates to estimate a standard error for each model coefficient. With this estimated standard error of each inferred model coefficient $\hat{\beta} \in \{\beta^{\hat{\mathrm{motif}}}, \beta^{\hat{\mathrm{AT}}}, \beta^{\hat{\mathrm{GC}}}\}$, we test whether $\hat{\beta} = 0$ by calculating the test statistic

$$T(\hat{\beta}) = \frac{\hat{\beta}}{\mathrm{se}(\hat{\beta})}$$

(12)

and comparing $T(\hat{\beta})$ to a $N(0,1)$ distribution to obtain each p-value. We consider the significance of each model coefficient using a Bonferroni-corrected threshold. To establish the threshold, we corrected for the total number of model coefficients being evaluated in the given model.

## Evaluating model coefficient variation in the context of SNPs

With the *motif* and *base-count-beyond* model, we are interested in quantifying variation in model coefficients in the context of genetic variations within the gene encoding the Artemis protein that were previously identified as being associated with increasing the extent of trimming (*Russell et al., 2022b*). Recall that we trained this model using the model covariate function given by (*Equation 2*). During model fitting, we estimated the model coefficients $\beta^{\hat{\mathrm{motif}}}$, $\beta^{\hat{\mathrm{AT}}}$, and $\beta^{\hat{\mathrm{GC}}}$ by maximizing the weighted likelihood function given by (*Equation 9*).

We have previously identified a set $X$ of SNPs within the gene encoding the Artemis protein that are significantly associated with increasing the extent of trimming (*Russell et al., 2022b*). For each SNP $x \in X$ and individual $i \in \{1, \ldots, I\}$, we measure the number of minor alleles in the genotype, $g_{ix} \in \{0, 1, 2\}$. We are interested in whether each of the inferred model coefficients

$\hat{\beta} \in \{\beta^{\hat{\mathtt{motif}}}, \beta^{\hat{\mathtt{AT}}}, \beta^{\hat{\mathtt{GC}}}\}$ vary in the context of genotype for each genetic variant $x \in X$. As such, for each SNP of interest, we can adapt the *1×2 motif + two-side base-count beyond* model covariate function to allow for genotype-specific variation of each model coefficient by incorporating additional interaction coefficients $\beta_x \in \{\beta_x^{\mathtt{motif}}, \beta_x^{\mathtt{AT}}, \beta_x^{\mathtt{GC}}\}$ to model the relationship between each model parameter and the SNP $x$ genotype. We can then estimate the coefficients of this new model, $\beta^{\hat{\mathtt{motif}}}$, $\beta^{\hat{\mathtt{AT}}}$, $\beta^{\hat{\mathtt{GC}}}$, $\beta_x^{\hat{\mathtt{motif}}}$, $\beta_x^{\hat{\mathtt{AT}}}$, and $\beta_x^{\hat{\mathtt{GC}}}$, as before by maximizing the weighted likelihood given by (*Equation 9*) using the adapted model covariate function. We can measure the significance of each of the model coefficients using the methods described in the previous section. Ultimately if a SNP-coefficient interaction term $\hat{\beta}_x \in \{\beta_x^{\hat{\mathtt{motif}}}, \beta_x^{\hat{\mathtt{AT}}}, \beta_x^{\hat{\mathtt{GC}}}\}$ is significant, we can conclude that the corresponding model coefficient $\hat{\beta}$ varies significantly in the context of the genotype of SNP $x \in X$. We use this same procedure to evaluate whether each model coefficient varies in the context of each SNP of interest.

## Acknowledgements

The authors thank David Schatz and Thayer Fisher for helpful discussions regarding this paper, as well as Duncan Ralph for processing the productive IGH sequence data from *Jaffe et al., 2022*, and Nathaniel Spisak, Thierry Mora, and Aleksandra Walczak for sharing preprocessed data from *Spisak et al., 2020*. The authors would also like to thank Fred Hutch scientific computing, supported by the National Institutes of Health award S10OD028685. This work was supported by the National Institutes of Health under awards R01 AI146028, R01 AI136514, and R35 GM141457. Dr. Matsen is an Investigator of the Howard Hughes Medical Institute (HHMI). This article is subject to HHMI's Open Access to Publications policy. HHMI lab heads have previously granted a nonexclusive CC BY 4.0 license to the public and a sublicensable license to HHMI in their research articles. Pursuant to those licenses, the author-accepted manuscript of this article can be made freely available under a CC BY 4.0 license immediately upon publication.

## Additional information

### Funding

| Funder | Grant reference number | Author |
| --- | --- | --- |
| National Institutes of Health | R01 AI146028 | Magdalena L Russell Philip Bradley Noah Simon Frederick A Matsen IV |
| National Institutes of Health | R01 AI136514 | Philip Bradley |
| National Institutes of Health | R35 GM141457 | Philip Bradley |
| Howard Hughes Medical Institute | Investigator | Frederick A Matsen IV |

The funders had no role in study design, data collection and interpretation, or the decision to submit the work for publication.

### Author contributions

Magdalena L Russell, Conceptualization, Software, Formal analysis, Validation, Investigation, Visualization, Methodology, Writing – original draft, Writing – review and editing; Noah Simon, Methodology, Writing – review and editing; Philip Bradley, Frederick A Matsen IV, Conceptualization, Supervision, Funding acquisition, Investigation, Methodology, Writing – review and editing

### Author ORCIDs

Magdalena L Russell http://orcid.org/0000-0002-1068-1968
Philip Bradley http://orcid.org/0000-0002-0224-6464
Frederick A Matsen IV, http://orcid.org/0000-0003-0607-6025

**Decision letter and Author response**
Decision letter https://doi.org/10.7554/eLife.85145.sa1
Author response https://doi.org/10.7554/eLife.85145.sa2

---

## Additional files

### Supplementary files
• MDAR checklist

### Data availability

The current manuscript is a computational study, so no data have been generated for this manuscript. Code is available on GitHub (copy archived at *Russell et al., 2022a*). Numerical data used to generate figures is available as source data for Figures 3, 4, 5, 6, and 7.

The following previously published datasets were used:

| Author(s) | Year | Dataset title | Dataset URL | Database and Identifier |
|---|---|---|---|---|
| Emerson RO, DeWitt WS, Vignali M, Gravley J, Osborne EJ, Desmarais C, Klinger M, Carlson CS, Hansen JA, Rieder M, Robins HS, Hu JK | 2017 | Immunosequencing identifies signatures of cytomegalovirus exposure history and HLA mediated effects on the T cell repertoire | https://doi.org/10.21417/B7001Z | ImmuneACCESS, 10.21417/B7001Z |
| Russell ML, Souquette A, Levine DM, Allen EK, Kuan G, Simon N, Balmaseda A, Gordon A, Thomas PG, Matsen FA, Bradley P | 2022 | Combining genotypes and T cell receptor distributions to infer genetic loci determining V(D)J recombination probabilities | https://www.ncbi.nlm.nih.gov/bioproject/PRJNA762269 | NCBI BioProject, PRJNA762269 |
| Robins H, Pearson O | 2015 | Normal Human PBMC Deep Sequencing TCRB versus TCRG comparison | https://clients.adaptivebiotech.com/pub/TCRB-TCRG-comparison | ImmuneACCESS, TCRB-TCRG-comparison |
| Briney B, Inderbitzin A, Joyce C, Burton DR | 2019 | Commonality despite exceptional diversity in the baseline human antibody repertoire | https://www.ncbi.nlm.nih.gov/bioproject/PRJNA406949 | NCBI BioProject, PRJNA406949 |
| Jaffe DB, Shahi P, Adams BA, Chrisman AM, Finnegan PM, Raman N, Royall AE, Tsai F, Vollbrecht T, Reyes DS, McDonnell WJ | 2022 | Functional antibodies exhibit light chain coherence | https://doi.org/10.25452/figshare.plus.20338177 | Figshare, 10.25452/figshare.plus.20338177 |
| Martin PJ, Levine DM, Storer BE, Nelson SC, Dong X, Hansen JA | 2020 | Recipient and donor genetic variants associated with mortality after allogeneic hematopoietic cell transplantation | https://www.ncbi.nlm.nih.gov/projects/gap/cgi-bin/study.cgi?study_id=phs001918.v1.p1 | NCBI dbGaP, phs001918 |

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

# Appendix 1

## Extended notation

**Appendix 1—table 1.** Summary of all notation used in our modeling.

| Variable | Description |
|---|---|
| General notation | |
| $I$ | Set of all individuals |
| $i$ | Index for an individual in the set $I$ of all individuals |
| $K_i$ | Total number of TCRs in the repertoire of individual $i$ |
| $k$ | Index of a sequence in the TCR repertoire of individual $i$ |
| $S$ | Random variable that represents the gene sequence |
| $\sigma$ | General notation for a gene-allele-group sequence oriented 5'-to-3' |
| $\sigma_V$ | V-gene-allele-group sequence ('top' strand oriented 5'-to-3') |
| $\sigma_J$ | J-gene-allele-group sequence ('bottom' strand oriented 5'-to-3') |
| $N$ | Random variable that represents the number of deleted nucleotides |
| $n$ | Number of deleted nucleotides from the 3'-side of a gene sequence |
| $L$ | Lower bound of 'reasonable' trimming amounts, we have defined $L = 2$ |
| $U$ | Upper bound of 'reasonable' trimming amounts, we have defined $U = 14$ |
| $C^{(i)}(\sigma)$ | The number of TCRs that use gene allele group $\sigma$ in the sampled repertoire of individual $i$ |
| $C^{(i)}(n, \sigma)$ | The number of TCRs that have gene allele group $\sigma$ and $n$ nucleotides deleted in the sampled repertoire of individual $i$ |
| $N'$ | Set of all 'reasonable' trimming amounts; $N' = \{2, \ldots 14\}$ |
| $P_{\mathrm{emp}}(N = n \mid S = \sigma, i)$ | Empirical conditional probability density function (**Equation 1**) |
| Motif parameter-specific notation | |
| $a$ | Non-negative integer value that represents the number of nucleotides 5' of the trimming site to be included in the 'trimming motif' |
| $b$ | Non-negative integer value that represents the number of nucleotides 3' of the trimming site to be included in the 'trimming motif' |
| $\{\sigma(n+j)\}_{j=-a}^{b-1}$ | 'Trimming motif' sequence (**Equation 13**) |
| $\beta_{js}^{\mathtt{motif}}$ | (Log) position weight matrix coefficient for trimming motif position $j \in \{-a, \ldots, b-1\}$ and nucleotide $s \in \{A, T, C, G\}$ |
| $\beta^{\mathtt{motif}}$ | Set of all *motif* coefficients $\beta_{js}^{\mathtt{motif}}$ for all positions $j \in \{-a, \ldots, b-1\}$ and nucleotide $s \in \{A, T, C, G\}$ |
| $f(n, \sigma; \beta^{\mathtt{motif}}, a, b)$ | *Motif*-specific covariate function (**Equation 14**) |
| Base-count-beyond parameter-specific notation | |
| $c$ | Non-negative integer value that represents the number of nucleotides 5' of the trimming site to be included in the 5' base-count-beyond the 'trimming motif' |
| $C^{\mathtt{AT}}(x)$ | Count of nucleotides that are A or T in an arbitrary sequence $x$ |
| $C^{\mathtt{GC}}(x)$ | Count of nucleotides that are G or C in an arbitrary sequence $x$ |

*Appendix 1—table 1 Continued on next page*

*Appendix 1—table 1 Continued*

| Variable | Description |
|---|---|
| $\text{seq}_5(n, \sigma, a, c)$ | The nucleotide sequence 5' of the trimming site, beyond the 'trimming motif' (**Equation 15**) |
| $\text{seq}_3(n, \sigma, b)$ | The nucleotide sequence 3' of the trimming site, beyond the 'trimming motif' (**Equation 16**) |
| $\beta_5^{\text{AT}}$ and $\beta_3^{\text{AT}}$ | *Base-count-beyond* model coefficients for the 5' and 3' sequence base-counts of A and T nucleotides beyond the trimming motif |
| $\beta^{\text{AT}}$ | Set of AT-*base-count-beyond* model coefficients (includes $\beta_5^{\text{AT}}$ and $\beta_3^{\text{AT}}$) |
| $\beta_5^{\text{GC}}$ and $\beta_3^{\text{GC}}$ | *Base-count-beyond* model coefficients for the 5' and 3' sequence base-counts of G and C nucleotides beyond the trimming motif |
| $\beta^{\text{GC}}$ | Set of GC-*base-count-beyond* model coefficients (includes $\beta_5^{\text{GC}}$ and $\beta_3^{\text{GC}}$) |
| $f(n, \sigma; \beta^{\text{AT}}, \beta^{\text{GC}}, a, b, c)$ | *Base-count-beyond*-specific covariate function (**Equation 14**) |
| DNA-shape parameter-specific notation | |
| $\text{seq}_{\text{expd}}(n, \sigma, a, b)$ | 'Expanded trimming sequence window' (**Equation 18**); consists of the 'trimming motif' sequence extended by 2 nucleotides in both the 5' and 3' direction |
| E | Nucleotide electrostatic potential |
| W | Nucleotide minor groove width |
| P | Nucleotide propeller twist |
| R | Di-nucleotide roll |
| H | Di-nucleotide helical twist |
| $\text{shape}^{\text{u}}(j, \text{seq}_{\text{expd}}(n, \sigma, a, b))$ | Measure of nucleotide shape $u \in \{\text{E}, \text{W}, \text{P}\}$ for the nucleotide at position $j \in \{-a, \dots, b-1\}$ within the 'expanded trimming sequence window' $\text{seq}_{\text{expd}}(n, \sigma, a, b)$ |
| $\text{shape}^{\text{v}}(d, \text{seq}_{\text{expd}}(n, \sigma, a, b))$ | Measure of di-nucleotide shape $v \in \{\text{R}, \text{H}\}$ for the di-nucleotide at position $d \in \{-a+1, \dots, b-1\}$ within the 'expanded trimming sequence window' $\text{seq}_{\text{expd}}(n, \sigma, a, b)$ |
| $\beta_{uj}^{\text{shape}}$ | DNA-shape coefficients for nucleotide shape type $u \in \{\text{E}, \text{W}, \text{P}\}$ and 'expanded trimming sequence window' nucleotide position $j \in \{-a, \dots, b-1\}$ |
| $\beta_{vd}^{\text{shape}}$ | DNA-shape coefficients for di-nucleotide shape type $v \in \{\text{R}, \text{H}\}$ and 'expanded trimming sequence window' di-nucleotide position $d \in \{-a+1, \dots, b-1\}$ |
| $\beta^{\text{shape}}$ | Set of all nucleotide and di-nucleotide DNA-shape coefficients |
| $f(n, \sigma; \beta^{\text{shape}}, a, b)$ | DNA-shape-specific covariate function (**Equation 19**) |
| Length parameter-specific notation | |
| $\beta^{\text{ldist}}$ | *Length* specific model coefficient |
| $f(n, \sigma; \beta^{\text{ldist}})$ | *Length*-specific covariate function |
| Modeling notation | |
| $f(n, \sigma; \beta^{\text{motif}}, \beta^{\text{AT}}, \beta^{\text{GC}}, a, b, c)$ | Example model covariate function including *motif* and *base-count-beyond* model parameters (**Equation 2**) |
| $P(n \mid \sigma; \beta^{\text{motif}}, \beta^{\text{AT}}, \beta^{\text{GC}}, a, b, c)$ | Conditional logit model formulation using the *motif* and *base-count-beyond* model covariate function (**Equation 3**) |

*Appendix 1—table 1 Continued on next page*

*Appendix 1—table 1 Continued*

| Variable | Description |
|---|---|
| $\log L(\beta^{\mathtt{motif}}, \beta^{\mathtt{AT}}, \beta^{\mathtt{GC}}, a, b, c)$ | Aggregated log-likelihood for the conditional logit model; this likelihood function is un-weighted (**Equation 4**) and gives every observation uniform treatment in the likelihood |
| $P_{\mathrm{samp}}(N = n, S = \sigma)$ | Sampling procedure for the construction of the expected likelihood |
| $\log L_{\mathrm{expected}}(\beta^{\mathtt{motif}}, \beta^{\mathtt{AT}}, \beta^{\mathtt{GC}}, a, b, c)$ | Expected log-likelihood for the conditional logit model; this likelihood function (**Equation 5**) weights each observation by its sampling probability, $P_{\mathrm{samp}}(N = n, S = \sigma)$ |
| $\log L_{\mathrm{emp}}(\beta^{\mathtt{motif}}, \beta^{\mathtt{AT}}, \beta^{\mathtt{GC}}, a, b, c)$ | Expected log-likelihood for the conditional logit model; this likelihood function (**Equation 7**) weights each observation by its sampling probability from the empirical joint PDF (**Equation 6**) |
| $P_{\mathrm{marg}}(\sigma)$ | Empirical average per-gene-allele-group frequency used in formulating a subject-independent gene sampling procedure (**Equation 8**) |
| $\log L_W(\beta^{\mathtt{motif}}, \beta^{\mathtt{AT}}, \beta^{\mathtt{GC}}, a, b, c)$ | Expected log-likelihood for the conditional logit model; this likelihood function (**Equation 9**) weights each observation using a subject-independent gene sampling procedure (**Equation 8**) |
| Model evaluation notation | |
| $\mathcal{M}$ | An arbitrary model trained on a specified training data set |
| $\mathbf{V}$ | Full V-gene data set |
| $\mathbf{J}$ | Full J-gene data set |
| $\mathbf{H}$ | Arbitrary held-out data set |
| $P(\mathbf{H})$ | Probability of the arbitrary held-out data set (**Equation 21**) |
| $\ell(\mathcal{M} \mid \mathbf{H})$ | Expected per-sequence conditional log loss (**Equation 11**) of a trained model $\mathcal{M}$ evaluated on a data set $\mathbf{H}$ |
| $E[\ell(\mathcal{M})]$ | Expected per-sequence conditional log loss across 20 random held-out data sets (**Equation 22**) |
| $\mathrm{RMSE}(\sigma, \mathcal{M}, \mathbf{V})$ | Per-gene mean squared error (**Equation 23**) for a gene $\sigma$ using a model $\mathcal{M}$ trained using the V-gene training data set $\mathbf{V}$ |
| Coefficient evaluation notation | |
| $T(\hat{\beta})$ | Test statistic (**Equation 12**) for evaluating the significance of a single inferred coefficient $\hat{\beta}$ |
| $X$ | Set of SNPs within the gene encoding the Artemis protein that were previously identified to be associated with increasing the extent of trimming (**Russell et al., 2022b**) |
| $g_{ix}$ | Number of minor alleles in the genotype of an individual $i \in I$ for SNP $x \in X$ |
| $\{\beta_x^{\mathtt{motif}}, \beta_x^{\mathtt{AT}}, \beta_x^{\mathtt{GC}}\}$ | Set of interaction coefficients between each model parameter and the SNP $x$ genotype |

## Appendix 2

### Extended parameter description

Defining the 'trimming motif' and position-weight-matrix weight for a given gene and trimming site

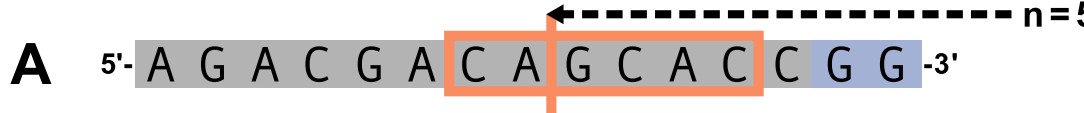

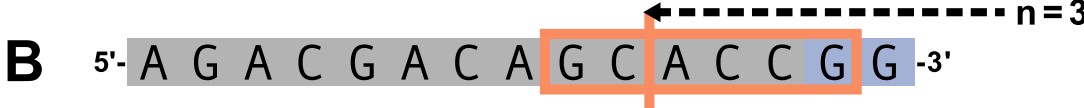

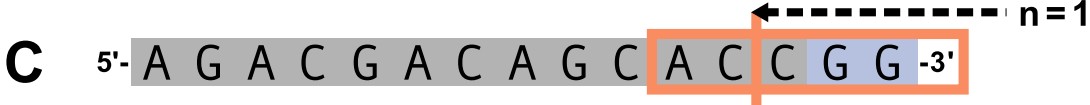

**Appendix 2—figure 1.** Summary of trimming motif parameters. Let $a = 2$ and $b = 4$. The 6-nucleotide trimming motif given by (**Equation 13**) is shown in the orange box and the trimming site is shown by the vertical orange line. An arbitrary gene sequence is highlighted in gray and the two possible P-nucleotides are highlighted in purple. (**A**) For $n = 5$, the 6-nucleotide trimming motif will not contain P-nucleotides. (**B**) For $n = 3$, the 6-nucleotide trimming motif will contain one P-nucleotide. (**C**) For $n = 1$, the trimming motif will contain two P-nucleotides and will be 'incomplete' (contain less than 6 nucleotides).

Existing probabilistic models of nucleotide trimming using repertoire sequencing data have shown that the local nucleotide context around the trimming site, which we refer to as the 'trimming motif,' do a surprisingly good job of predicting the distribution of trimming lengths for a variety of genes (**Murugan et al., 2012**). This simple PWM model uses a trimming motif containing 2 nucleotides 5' of the trimming site and 4 nucleotides 3' of the trimming site to predict the probability of trimming at that site. In practice, we can define the trimming motif to be any size. Let $a$ and $b$ be non-negative integer values that represent the number of nucleotides 5' and 3' of the trimming site, respectively. Together, these $a + b$ nucleotides will compose the trimming motif. For a gene-allele-group sequence $\sigma$ and a number of deleted nucleotides $n$, let $\sigma(n + j)$ represent the nucleotide identity at the trimming motif position $j \in \{-a, \ldots, b - 1\}$ where positions $j < 0$ represent motif positions 5' of the trimming site and positions $j \geq 0$ represent motif positions 3' of the trimming site. As such, the trimming motif sequence is given by

$$\{\sigma(n + j)\}_{j=-a}^{b-1}. \tag{13}$$

Depending on $n$, this trimming motif may or may not include P-nucleotides. For example, for $n \geq b$, the $b$ 3' trimming motif nucleotides will include the $b$ deleted gene sequence nucleotides 3' of the trimming site (and no P-nucleotides) (**Appendix 2—figure 1A**). Since we are assuming that the initial hairpin nick occurs at the +2 position, there will be two P-nucleotides present in the 5'-to-3' gene sequence. For $b - 2 \leq n < b$, where the 2 represents the total P-nucleotide count in the full sequence, P-nucleotides will be included in the trimming motif sequence. Specifically, the $b$ total 3' trimming motif nucleotides will include $b - n$ P-nucleotides and $n$ deleted gene sequence nucleotides (**Appendix 2—figure 1B, C**). Likewise, as a result of the +2 hairpin nick position assumption, TCRs that have $n < b - 2$ will not have a full, $(a + b)$-length nucleotide trimming motif (**Appendix 2—figure 1C**). For these 'off-the-end' motif cases, we assign zero influence to the missing nucleotides during model fitting.

With this trimming motif, let $\beta_{js}^{\texttt{motif}}$ be a (log) position-weight-matrix coefficient for trimming motif position $j \in \{-a, \ldots, b-1\}$ and nucleotide $s \in \{A, T, C, G\}$. We can define an un-normalized position-weight-matrix weight

$$f(n, \sigma; \beta^{\texttt{motif}}, a, b) := \sum_{j=-a}^{b-1} \beta_{j\sigma(n+j)}^{\texttt{motif}} \tag{14}$$

that will serve as a *motif*-specific model covariate function in subsequent modeling. As described above, since we are considering 'off-the-end' motif cases, $\sigma(n+j)$ represent the nucleotide identity at sequence position $j$ where positions $j < 0$ represent sequence positions 5' of the trimming site and positions $j \geq 0$ represent sequence positions 3' of the trimming site.

## AT and GC base-count-beyond the trimming motif

For an arbitrary sequence $x$, we can count the number of AT and GC nucleotides within the sequence as

$$C^{\texttt{AT}}(x) = C^{\texttt{A}}(x) + C^{\texttt{T}}(x)$$

and

$$C^{\texttt{GC}}(x) = C^{\texttt{G}}(x) + C^{\texttt{C}}(x),$$

respectively.

Because the count of AT or GC nucleotides within the sequences 5' and 3' of the trimming site may influence the probability of trimming differently, we will calculate the counts separately. We will not include nucleotides that were already included in the *motif* parameterization. As above, for a gene-allele-group sequence $\sigma$ and a number of deleted nucleotides $n$, let $\sigma(n+j)$ represent the nucleotide identity at sequence position $j$ where positions $j < 0$ represent sequence positions 5' of the trimming site and positions $j \geq 0$ represent sequence positions 3' of the trimming site. Let $c$ be a non-negative integer value that represents the number of nucleotides 5' of the trimming site that will be included in the 5'-nucleotide counts (*Appendix 2—figure 2*). Recall that $a$ is a non-negative integer value that represents the number of nucleotides 5' of the trimming site that are included in the 'trimming motif' described in the previous section. As such, the nucleotide sequence 5' of the trimming site, beyond the 'trimming motif,' is given by

$$\text{seq}_5(n, \sigma, a, c) = \{\sigma(n+j)\}_{j=(a+1)}^{(a+c)}. \tag{15}$$

Within this sequence $\text{seq}_5(n, \sigma, a, c)$, we can count the number of AT and GC nucleotides as

$$C^{\texttt{AT}}(\text{seq}_5(n, \sigma, a, c)) = C^{\texttt{A}}(\text{seq}_5(n, \sigma, a, c)) + C^{\texttt{T}}(\text{seq}_5(n, \sigma, a, c))$$

and

$$C^{\texttt{GC}}(\text{seq}_5(n, \sigma, a, c)) = C^{\texttt{G}}(\text{seq}_5(n, \sigma, a, c)) + C^{\texttt{C}}(\text{seq}_5(n, \sigma, a, c)),$$

respectively

To count the number of AT and GC nucleotides in the sequence 3' of the trimming site, we will include all nucleotides located 3' of the trimming site that are beyond the 'trimming motif.' However, because we are interested in using GC nucleotide content in both directions of the wider sequence as a proxy for the capacity for sequence-breathing and since sequence-breathing is only relevant for nucleotides that are paired, we will not include the nucleotides within the 3' single-stranded-overhang when counting 3' AT and GC nucleotides (*Appendix 2—figure 2*). Since we are assuming that the initial hairpin nick occurs at the +2 position leading to a 4-nucleotide-long 3' single-stranded-overhang, for $n > 2$, the nucleotide sequence 3' of the trimming site, beyond the 'trimming motif,' is given by

$$\text{seq}_3(n, \sigma, b) = \begin{cases} \{\sigma(n+j)\}_{j=3-n}^{-b} & \text{if } (n-3) \geq b \\ \{\} & \text{if } (n-3) < b \end{cases} \tag{16}$$

where $b$ is a non-negative integer value that represents the number of nucleotides 3′ of the trimming site that are included in the 'trimming motif' described in the previous section. For $(n-3) < b$, all nucleotides 3′ of the trimming site are considered single-stranded and, thus, no nucleotides will be included in the sequence used to calculate the AT and GC base-counts (*Appendix 2—figure 2C*). Within this sequence $\text{seq}_3(n, \sigma, b)$, we can count the number of AT and GC nucleotides as

$$C^{\text{AT}}(\text{seq}_3(n, \sigma, b)) = C^{\text{A}}(\text{seq}_3(n, \sigma, b)) + C^{\text{T}}(\text{seq}_3(n, \sigma, b))$$

and

$$C^{\text{GC}}(\text{seq}_3(n, \sigma, b)) = C^{\text{G}}(\text{seq}_3(n, \sigma, b)) + C^{\text{C}}(\text{seq}_3(n, \sigma, b)),$$

respectively. As defined, these GC and AT base-counts for the 3′ sequence are dependent on sequence length and provide a parameterization of both GC nucleotide content in both directions of the wider sequence and length.

With these 5′ and 3′ base counts, we can define $\beta_5^{\text{AT}}$, $\beta_3^{\text{AT}}$, $\beta_5^{\text{GC}}$, and $\beta_3^{\text{GC}}$ to be *base-count-beyond* model coefficients for 5′ and 3′ sequence base-counts of AT and GC beyond the 'trimming motif,' respectively. With these coefficients, we can define a *base-count-beyond* covariate function for each trimming site $n$ and gene $\sigma$:

$$f(n, \sigma; \boldsymbol{\beta}^{\text{AT}}, \boldsymbol{\beta}^{\text{GC}}, a, b, c) := \beta_5^{\text{AT}} \cdot C^{\text{AT}}(\text{seq}_5(n, \sigma, a, c)) + \beta_3^{\text{AT}} \cdot C^{\text{AT}}(\text{seq}_3(n, \sigma, b))$$
$$+ \beta_5^{\text{GC}} \cdot C^{\text{GC}}(\text{seq}_5(n, \sigma, a, c)) + \beta_3^{\text{GC}} \cdot C^{\text{GC}}(\text{seq}_3(n, \sigma, b)). \tag{17}$$

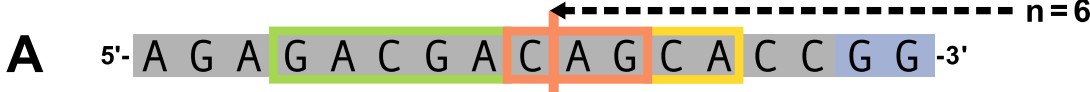

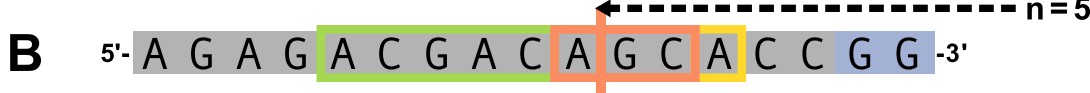

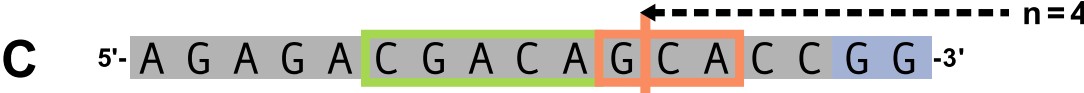

**Appendix 2—figure 2.** Summary of base-count parameters. Let $a = 1$, $b = 2$, and $c = 5$. An arbitrary gene sequence is highlighted in gray and the two possible P-nucleotides are highlighted in purple. The trimming site is shown by the vertical orange line and the 'trimming motif,' as defined in (*Equation 13*), is shown by the orange box. The $c$ nucleotides included in the count of AT and GC nucleotides 5′ of the trimming site, beyond the 'trimming motif,' are expressed by (*Equation 15*) and are shown in the green box. The nucleotides included in the count of AT and GC nucleotides 3′ of the trimming site, beyond the 'trimming motif,' are expressed by (*Equation 16*) and are shown in the yellow box. As described in the text, we are assuming that the initial hairpin nick occurs at the +2 position leading to a 4-nucleotide-long 3′ single-stranded-overhang. We exclude these single-stranded nucleotides in the 3′-base-count-beyond sequence. In this figure, the 4 nucleotides nearest to the 3′ side of each sequence (this includes the two P-nucleotides and the two 3′-most gene sequence nucleotides) are considered single-stranded and will not be included in the 3′-base-count-beyond sequence. (**A**) For $n = 6$, 2 nucleotides 3′ of the trimming site will be used in the 3′ sequence base-counts. (**B**) For $n = 5$, 1 nucleotide 3′ of the trimming site will be used in the 3′ sequence base-counts. (**C**) For $n = 4$, all nucleotides 3′ of the trimming site are considered single-stranded and, thus, no nucleotides will be used for the 3′ sequence base-counts.

## DNA-shape around the trimming site

Methods have been previously developed to estimate DNA-shape features at a single-nucleotide position using the sequence context of 2 neighboring nucleotides on both sides of the nucleotide of interest (*Zhou et al., 2013*; *Chiu et al., 2016*). As such, these methods use a sliding-pentamer model, centered at each nucleotide of interest, to derive the structural features of nucleotides within a sequence window of any length. These structural features include estimations of electrostatic potential (E), minor groove width (W), and propeller twist (P) for each nucleotide in the sequence window and estimations of roll (R) and helical twist (H) for each di-nucleotide pair in the sequence window. For simplicity, we will use the term 'DNA-shape parameters' to refer to all five of these structural features.

For our purposes, we can define a 'trimming sequence window' of size $a + b$, as introduced in the 'trimming motif' section with (*Equation 13*), where $a$ and $b$ are non-negative integer values that represent the number of nucleotides 5' and 3' of the trimming site, respectively. In order to estimate the DNA-shape for all nucleotides within this window, we will expand the 'trimming sequence window' by 2 nucleotides on both sides such that there are $a + 2$ nucleotides 5' and $b + 2$ nucleotides 3' of the trimming site included in an 'expanded trimming sequence window.' For a gene-allele-group sequence $\sigma$ and a number of deleted nucleotides $n$, let $\sigma(n + j)$ represent the nucleotide identity at the 'expanded trimming sequence window' position $j \in \{-(a + 2), \ldots, (b + 2) - 1\}$ where positions $j < 0$ represent expanded trimming sequence window positions 5' of the trimming site and positions $j \geq 0$ represent expanded trimming sequence window positions 3' of the trimming site. As such, the expanded trimming sequence window is given by

$$\mathrm{seq}_{\mathrm{expd}}(n, \sigma, a, b) := \{\sigma(n + j)\}_{j=-(a+2)}^{(b+2)-1}. \tag{18}$$

Depending on $n$, this expanded trimming sequence window may or may not include P-nucleotides. For example, for $n \geq (b + 2)$, the $(b + 2)$ 3' expanded trimming sequence window nucleotides will include the $(b + 2)$ deleted gene sequence nucleotides 3' of the trimming site (and no P-nucleotides) (*Appendix 2—figure 3A*). For $b \leq n < b + 2$, the $(b + 2)$ 3' expanded trimming sequence window nucleotides will include $(b + 2) - n$ P-nucleotides and $n$ deleted gene sequence nucleotides (*Appendix 2—figure 3B*). Since we are assuming that the initial hairpin nick occurs at the +2 position, TCRs that have $n < b$ will not have a full, $(a + b + 4)$-length nucleotide expanded trimming sequence window (*Appendix 2—figure 3C*). The sliding-pentamer model (*Zhou et al., 2013*; *Chiu et al., 2016*) requires a full pentamer for estimating the DNA-shape of each base of interest, and, thus, for these 'off-the-end' expanded trimming sequence window cases, we cannot estimate DNA-shape parameters for all nucleotides within the trimming sequence window. As such, when estimating DNA-shape parameters, we must choose $b$ such that $b \leq n$ for all trimming lengths $n$ in the data set.

For each nucleotide position $j \in \{-a, \ldots, b - 1\}$ within the expanded trimming sequence window $\mathrm{seq}_{\mathrm{expd}}(n, \sigma, a, b)$, we can estimate the nucleotide electrostatic potential, $\mathrm{shape}^{\mathrm{E}}(j, \mathrm{seq}_{\mathrm{expd}}(n, \sigma, a, b))$, minor groove width, $\mathrm{shape}^{\mathrm{W}}(j, \mathrm{seq}_{\mathrm{expd}}(n, \sigma, a, b))$, and propeller twist, $\mathrm{shape}^{\mathrm{P}}(j, \mathrm{seq}_{\mathrm{expd}}(n, \sigma, a, b))$. We then standardize the estimated values for each shape type. We can define $\beta_{uj}^{\mathrm{shape}}$ to be a nucleotide shape model coefficient for nucleotide shape type $u \in \{\mathrm{E}, \mathrm{W}, \mathrm{P}\}$ and trimming sequence window nucleotide position $j \in \{-a, \ldots, b - 1\}$. Let $d \in \{-a + 1, \ldots, b - 1\}$ be the location of each di-nucleotide in the trimming sequence window such that $d = 0$ represents the location of the trimming site, $d < 0$ represents di-nucleotide positions 5' of the trimming site, and $d > 0$ represents di-nucleotide positions 3' of the trimming site. For each di-nucleotide $d \in \{-a + 1, \ldots, b - 1\}$ within the expanded trimming sequence window $\mathrm{seq}_{\mathrm{expd}}(n, \sigma, a, b)$, we can estimate the di-nucleotide roll, $\mathrm{shape}^{\mathrm{R}}(d, \mathrm{seq}_{\mathrm{expd}}(n, \sigma, a, b))$ and helical twist, $\mathrm{shape}^{\mathrm{H}}(d, \mathrm{seq}_{\mathrm{expd}}(n, \sigma, a, b))$. As above, we then standardize the estimated values for each di-nucleotide shape type. We can define $\beta_{vd}^{\mathrm{shape}}$ to be a di-nucleotide shape model coefficient for di-nucleotide shape type $v \in \{\mathrm{R}, \mathrm{H}\}$ and trimming sequence window di-nucleotide position $d \in \{-a + 1, \ldots, b - 1\}$. We use the R package DNAshapeR (*Chiu et al., 2016*) to estimate these DNA-shape parameters for each trimming sequence window. With these standardized DNA-shape estimates, we can define a DNA-shape covariate function for each trimming site $n$ and gene $\sigma$

$$f(n, \sigma; \beta^{\text{shape}}, a, b) := \sum_{j=-a}^{b-1} \sum_{u \in \{\text{E,W,P}\}} \beta_{uj}^{\text{shape}} \cdot \text{shape}^u(j, \text{seq}_{\text{expd}}(n, \sigma, a, b))$$

$$+ \sum_{d=-a+1}^{b-1} \sum_{v \in \{\text{R,H}\}} \beta_{vd}^{\text{shape}} \cdot \text{shape}^v(d, \text{seq}_{\text{expd}}(n, \sigma, a, b)). \tag{19}$$

**Appendix 2—figure 3.** Summary of DNA-shape parameters. Let $a = 1$ and $b = 2$. The 3-nucleotide trimming sequence window is shown in the orange box and the trimming site is shown by the vertical orange line. The 7-nucleotide expanded trimming sequence window is represented by the pink boxes in addition to the original trimming sequence window orange box. An arbitrary gene sequence is highlighted in gray and the two possible P-nucleotides are highlighted in purple. (**A**) For $n = 5$, both the 7-nucleotide expanded trimming sequence window and the original 3-nucleotide trimming sequence window will not contain P-nucleotides. (**B**) For $n = 3$, the 7-nucleotide expanded trimming sequence window will contain one P-nucleotide and the original 3-nucleotide trimming sequence window will not contain P-nucleotides. (**C**) For $n = 1$, the 7-nucleotide expanded trimming sequence window will be 'incomplete' (contain less than 7 nucleotides), and thus, will be invalid for estimating DNA-shape for the nucleotides within the original trimming sequence window.

## Length

We can think of the trimming amount $n$ as a measure of the sequence-independent length from the end of the gene for each gene and trimming site, and define $\beta^{\text{ldist}}$ to be a *length* model coefficient. As such, we can define a length covariate function for each trimming site $n$

$$f(n, \sigma; \beta^{\text{ldist}}) := \beta^{\text{ldist}} \cdot n. \tag{20}$$

## Appendix 3

### Extended model validation methods

Calculating the expected per-sequence conditional log loss across the full V-gene training data set

With the full V-gene training set, we can train each model of interest as described above in (**Equation 10**) to obtain a trained model $\mathcal{M}$. After this model fitting, we can calculate the expected per-sequence conditional log loss of the model, $\mathcal{M}$, for the full V-gene training data set, $\mathbf{V}$, using the procedure described above in (**Equation 11**). Here, we use the full V-gene data set as both the training data set and the testing data set. Models that have lower expected per-sequence conditional log loss on the V-gene training data set will indicate that the model has a better fit. Model evaluation using held-out testing sets, as described below, is required for evaluating model generalizability.

Calculating the expected per-sequence conditional log loss across held-out samples

Because our goal is to learn a model that is gene-agnostic, we will evaluate the performance and generalizability of each model by calculating the expected per-sequence conditional log loss using many different held-out data sets. A model that is generalizable across many genes will perform well and have a good fit across all held-out samples despite their varying gene compositions. To test this, we will create each random, held-out sample from the original training data set by cluster-sampling all observations from V-gene allele groups, $\sigma_V$, uniformly at random. We will refer to each random, held-out sample as the 'held-out testing set.' Let $G$ be the total number of unique V-gene allele groups in the original data set. Let $G_{\text{test}} = \text{Round}(0.3 \cdot G)$ be an integer which represents the number of unique genes included in each 'held-out testing set.' As such, we can sample each gene $\sigma_V$ with probability

$$P_{\text{sample}}(S = \sigma_V) := \frac{1}{G}$$

such that the probability of each 'held-out testing set' H is given by

$$
\begin{aligned}
P(\mathbf{H}) &= \prod_{\sigma_V=1}^{G_{\text{test}}} P_{\text{sample}}(S = \sigma_V) \\
&= \prod_{\sigma_V=1}^{G_{\text{test}}} \frac{1}{G}.
\end{aligned}
\tag{21}
$$

The remaining genes not sampled as part of the 'held-out testing set' $\mathbf{H}$ will compose the 'training set' $\mathbf{T}$. Using this 'training set,' we can train each model of interest as described above in (**Equation 10**). After this model training, we can calculate the expected per-sequence conditional log loss of the model, $\mathcal{M}$, for the 'held-out testing set,' $\mathbf{H}$, as described above in (**Equation 11**). To achieve an unbiased estimate of the model performance, we will repeat the above procedure across 20 unique held-out testing sets and calculate the expected per-sequence conditional log loss across all samples. As such, the expected per-sequence conditional log loss across these random samples is given by

$$E[\ell(\mathcal{M})] = \sum_{\mathbf{H}=1}^{20} P(\mathbf{H}) \cdot \ell(\mathcal{M} \mid \mathbf{H}).
\tag{22}$$

We use the same, unique held-out testing sets to calculate the expected per-sequence conditional log loss of each model of interest, and thus, we can compare model fit and generalizability by directly comparing the expected per-sequence conditional log loss of each model. Models that have lower expected per-sequence conditional log loss will indicate that the model is a better fit and is more generalizable across genes.

## Calculating the expected per-sequence conditional log loss across held-out samples of the 'most-different' V-genes

While the previously described procedure for evaluating the expected per-sequence conditional log loss across held-out samples of the V-gene data set provided a metric for evaluating model generalizability across different gene sets, we were interested in evaluating model performance for groups of genes which were considered 'most-different' sequence-wise. Many of the germline V-gene sequences are quite similar, however, there are subgroups of these sequences which share unique sequence traits. We can characterize these 'most-different' V-genes by either using only the 'terminal' V-gene sequences (e.g. that last 24 nucleotides of each sequence which is directly parameterized in the models) or using the entire V-gene sequences.

To define the 'most-different' V-gene allele group using the 'terminal' V-gene sequences, we first calculate the pairwise hamming distance between each gene-allele-group pair. We then use hierarchical clustering to cluster V-gene allele groups based on their pairwise hamming distances (*Appendix 3—figure 1A*). The cluster that has the smallest average pairwise hamming distance within the cluster and the largest average pairwise hamming distance outside of the cluster is defined to be the 'most-different' V-gene-allele-group cluster. To define the 'most-different' V-gene allele group using the entire V-gene sequences, we first align all gene sequences using the DECIPHER package in R. Using these aligned sequences, we can then proceed with the same procedure as described for the 'terminal' V-gene sequences to define the 'most-different' V-gene allele group (*Appendix 3—figure 1B*).

Once we have defined a cluster of the 'most-different' V-gene allele groups, using either the 'terminal' V-gene sequences or the full sequences, we can define a held-out testing data set **H** containing all data observations from the V-gene allele groups within this 'most-different' V-gene-allele-group cluster. All data observations from the remaining gene allele groups that were not defined to be part of the 'most-different' cluster will compose the 'training set' **T**. Using this 'training set,' we can train each model of interest as described above in (*Equation 10*). After this model training, we can calculate the expected per-sequence conditional log loss of the model, $\mathcal{M}$, for the 'held-out testing set,' **H**, as described above in (*Equation 11*). Models that have lower expected per-sequence conditional log loss will indicate better fit and generalizability across even the 'most-different' genes. We can repeat this process for other V-gene-allele-group clusters (e.g. the 'second-most-different' V-gene-allele-group cluster) as desired.

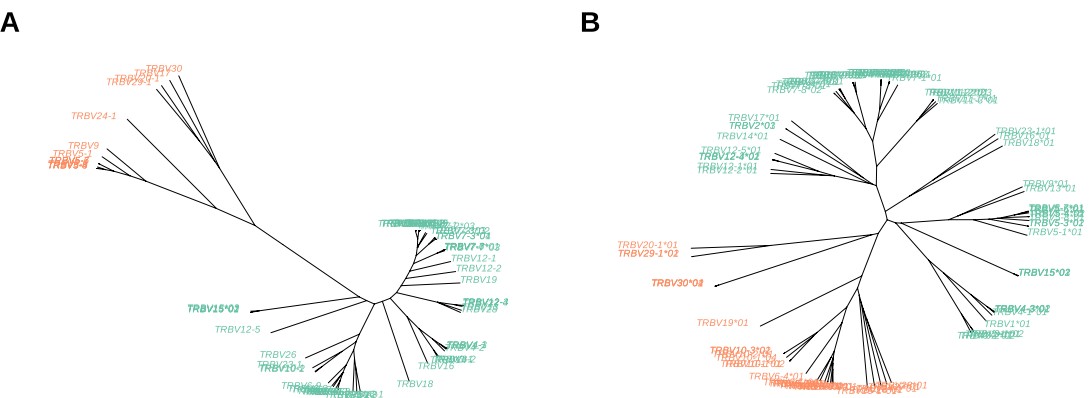

**Appendix 3—figure 1.** Un-rooted trees of 'terminal' V-gene sequences (**A**) and full-length V-gene sequences (**B**) derived from hierarchical clustering. Tips are colored according to cluster membership. The tips corresponding to the 'most-different' group within each tree are colored in orange.

## Calculating the expected per-sequence conditional log loss across the full J-gene data set

With the full V-gene training set, we can train each model of interest as described above in (*Equation 10*) to obtain a trained model $\mathcal{M}$. After this model fitting, we can calculate the expected per-sequence conditional log loss of the model, $\mathcal{M}$, for the full J-gene training data set, **J**, using the procedure described above in (*Equation 11*). Here, we use the full V-gene data set as the training

data set and the full J-gene data set as the testing data set. Models that have lower expected per-sequence conditional log loss on the J-gene data set will indicate that the model is a better fit and is more generalizable.

## Evaluating TCRβ V-gene trimming models using the expected per-sequence conditional log loss across testing data sets

To validate the performance of each model, we worked with TCRα- and TCRβ-immunosequencing data representing 150 individuals, TCRγ-immunosequencing data representing 23 individuals, and IGH-immunosequencing data representing 9 individuals from three independent validation cohorts (described above). With these data, we used the model coefficients from the previous TCRβ V-gene training run ('frozen' in git commit 093610a on our repository) and then compute the expected per-sequence conditional log loss of the model using each independent validation data set of interest. Models that have low expected per-sequence conditional log loss across all testing data sets will indicate that the model is more generalizable and less overfit to the training data. We validated each model using V- and J-gene sequences separately.

## Appendix 4

### Extended experimental analyses

### Exploring the gene specificity of the 'trimming motif'

To evaluate the specificity of the *motif* coefficients across different genes, we can compare the per-gene model predictions for the *motif* and *base-count beyond* model to a model that only contains *base-count beyond* parameters. To do this, we first use the entire V-gene data set $\mathbf{V}$ to train both the *motif* and *base-count beyond* model as before in (*Equation 10*) and a model that contains only *base-count beyond* parameters (and no motif parameters). We can then use these models to predict the probability of trimming each possible trimming amount, $2 \leq n \leq 14$, for each gene-allele-group sequence $\sigma$. For each of these models, we can then calculate the per-gene root mean squared error, RMSE, for each gene $\sigma$ such that

$$\mathrm{RMSE}(\sigma, \mathcal{M}, \mathbf{V}) = \sqrt{\frac{\sum_{i=1}^{I} \sum_{n=2}^{14} \left(P_{\mathrm{emp}}(n \mid \sigma, i) - P(n \mid \sigma; \mathcal{M})\right)^2}{|I|}} \tag{23}$$

where $\mathcal{M}$ is a model trained using the V-gene training data set $\mathbf{V}$, $I$ is the set of all individuals in the data set, $|I|$ is the length of the set of individuals $I$, $P_{\mathrm{emp}}(n \mid \sigma, i)$ is the empirical conditional PDF given by (*Equation 1*) for trimming length $n$, gene $\sigma$, and individual $i \in I$, and $P(n \mid \sigma; \mathcal{M})$ is the predicted trimming probability from a specified model $\mathcal{M}$. We can then compare this per-gene root mean squared error for the model trained using both *motif* and *base-count beyond* parameters with a model trained using just *base-count beyond* parameters.

### Sensitivity analysis for hairpin nick position

For our modeling, we assume that the initial hairpin nick occurs at the +2 position and will create two P-nucleotides at the end of the 5'-to-3' gene sequence. Assuming a different hairpin nick position would incorporate a different number of P-nucleotides at the end of the gene sequence (*Appendix 4—figure 1*). While the hairpins are assumed to be nicked at the +2 position most frequently (*Ma et al., 2002*; *Lu et al., 2007*), we wanted to test the sensitivity of our models to this hairpin nick position assumption. To do this, we assumed each of the other possible hairpin opening positions (e.g. −2,−1, 0,+1,+3) one-at-a-time and appended the appropriate number of associated of P-nucleotides given the assumed hairpin nick position to the 3'-end of each V-gene-allele-group sequence in the data set. With each of these hairpin position data sets, we re-trained the *motif* and *base-count beyond* model as before in (*Equation 10*) and calculate the expected per-sequence conditional log loss of the model using (*Equation 11*). We can compare these expected per-sequence conditional log losses to evaluate the sensitivity of the model to the +2 hairpin nick assumption.

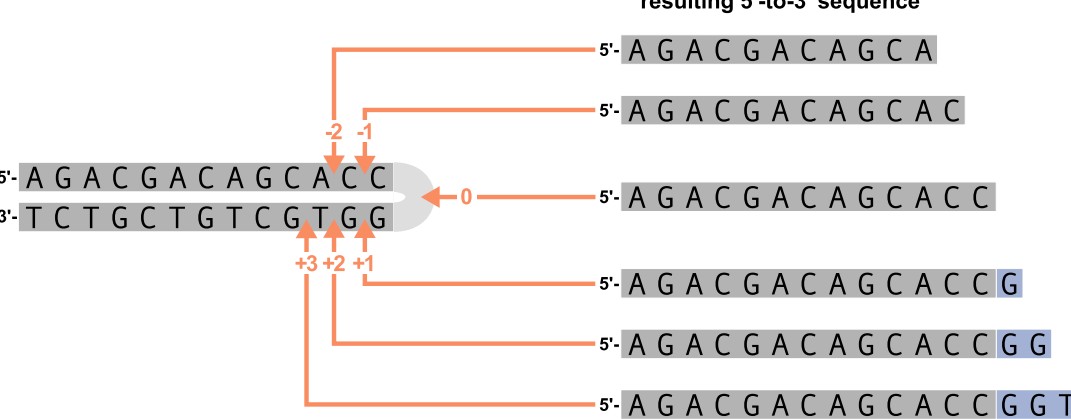

**Appendix 4—figure 1.** An arbitrary DNA hairpin can be nicked opened at various positions near the hairpin (left figure). Hairpin nick position 0 refers to a nick at the tip of the hairpin, position –1 refers to a nick before the

*Appendix 4—figure 1 continued on next page*

*Appendix 4—figure 1 continued*

last nucleotide on the 5' strand, position +1 refers to a nick before the last nucleotide on the 3' strand, etc. The resulting 5'-to-3' sequences from the various nick positions for the arbitrary gene sequence are shown on the right. Nucleotides originating from the 5' strand of the DNA hairpin are highlighted in gray and P-nucleotides (originating from the 3' strand of the DNA hairpin) are highlighted in purple. The various hairpin nick positions lead to 5'-to-3' sequences that contain different amounts of P-nucleotides. Hairpin nick positions $gt_0$ lead to 5'-to-3' sequences that contain P-nucleotides, nick positions equal to zero lead to 5'-to-3' sequences without P-nucleotides, and nick positions < 0 lead to 5'-to-3' sequences without P-nucleotides and with portions of the original 5' DNA hairpin strand removed.

## Evaluating the weight of the *1×2 motif* and *two-side base-count beyond* model terms across data sets

For each testing data set, we can measure the weight of the *1×2 motif* and *two-side base-count beyond* model terms within the full *1×2 motif + two-side base-count beyond* model. Recall that we trained the full *1×2 motif + two-side base-count beyond* model using the model covariate function given by (*Equation 2*)

$$f(n, \sigma_{\mathrm{V}}; \beta^{\mathtt{motif}}, \beta^{\mathrm{AT}}, \beta^{\mathrm{GC}}, a, b, c) := f(n, \sigma_{\mathrm{V}}; \beta^{\mathtt{motif}}, a, b) + f(n, \sigma_{\mathrm{V}}; \beta^{\mathrm{AT}}, \beta^{\mathrm{GC}}, a, b, c)$$

where $n$ represents the number of trimmed nucleotides, $\sigma_{\mathrm{V}}$ represents the V-gene-allele-group sequence, $\beta^{\mathtt{motif}}$ represents *motif*-specific parameter coefficients, $\beta^{\mathrm{AT}}$ and $\beta^{\mathrm{GC}}$ represent *base-count-beyond*-specific parameter coefficients, $a$ and $b$ are non-negative integer values that represent the number of nucleotides 5' and 3' of the trimming site, respectively, that are included in the 'trimming motif', and $c$ represents the number of nucleotides 5' of the trimming site to be included in the base-count. As such, for each training data set, we can use the inferred coefficients, $\beta^{\hat{\mathtt{motif}}}$, $\hat{\beta}^{\mathrm{AT}}$, and $\hat{\beta}^{\mathrm{GC}}$, from a previous training run and define a new two-parameter model containing a scale coefficient for the *1×2 motif* terms and a second scale coefficient for the *two-side base-count beyond* terms. The covariate function for this new model is given by

$$f(n, \sigma_{\mathrm{V}}; \beta^{\hat{\mathtt{motif}}}, \hat{\beta}^{\mathrm{AT}}, \hat{\beta}^{\mathrm{GC}}, \alpha_{\mathtt{motif}}, \alpha_{\mathtt{count}}, a, b, c) := \alpha_{\mathtt{motif}} \cdot f(n, \sigma_{\mathrm{V}}; \beta^{\mathtt{motif}}, a, b)$$
$$+ \alpha_{\mathtt{count}} \cdot f(n, \sigma_{\mathrm{V}}; \beta^{\mathrm{AT}}, \beta^{\mathrm{GC}}, a, b, c)$$

where $\alpha_{\mathtt{motif}}$ is the scale coefficient for the *1×2 motif* terms and $\alpha_{\mathtt{count}}$ is the scale coefficient for the *two-side base-count beyond* terms. We can then train this new model as described previously for each data set of interest and compare the inferred scale coefficients.

