## [Editor Report]

Russell et al. study and reveal compelling evidence for potential sequence-based factors that may drive VDJ trimming, a mechanism involved in VDJ recombination that shapes adaptive immune repertoire generation. The work is based on a rigorous statistical comparison of logistic regression models to reveal the role and function of cutting enzymes in shaping T- and B-cell receptor diversity which could provide fundamental new insights into these processes.

---

## [Decision Letter]

**Decision letter after peer review:**

Thank you for submitting your article "Statistical inference reveals the role of length, breathing, and nucleotide identity in V(D)J nucleotide trimming" for consideration by *eLife*. Your article has been reviewed by 2 peer reviewers, and the evaluation has been overseen by a Reviewing Editor and Betty Diamond as the Senior Editor. The following individual involved in the review of your submission has agreed to reveal their identity: Thierry Mora (Reviewer #2).

Your manuscript addresses an important topic and interesting approach to foster our understanding of processes relevant to immune repertoire generation. However, there were several aspects identified during the review process that would require major revisions to support or adapt the claims made. In particular, this affects the following essential revisions:

Essential revisions:

1) A rephrasing or additional support for the claim to provide mechanistic insights which seem to be overstated. Based on the fact that only statistical models are used, there is currently no real indication of mechanistic or quantitative insight into the involved processes.

2) An extensive restructuring and rewriting of the manuscript to clarify the focus of the paper (see comments of reviewer 3, e.g. regarding the extended methods section)

3) Improved explanation of the model, as well as several details concerning the statistical analyses

*Reviewer #1 (Recommendations for the authors):*

Really great work and very interesting. My recommendations are mostly questions and some suggestions.

– The text is really dense (especially when you talk about and compare all the different models tested). Can that somehow be shortened and made more concise?

– Maybe it's mentioned somewhere, but how many sequences were in train and test datasets? Have you performed subsampling studies to understand at what sequence number your models become stable?

– You only use unselected sequences. If you had used productive sequences, would the results have been dramatically different?

– You are somehow splitting the datasets by V gene distance. How would the results have looked like with a random train/test split?

– Figure 3a, 4a what's the statistical significance between curves? I rarely see error bars (if at all) in any of the figures. To what extent are your results dependent on just sampling once?

– You use the term mechanism a lot. The title is also quite strongly worded ("reveal role of"). To what extent is this justified given that you "only" perform statistical modeling and no experimental investigations? To what extent are you sure that your models are really a reflection of biology (causal)?

*Reviewer #2 (Recommendations for the authors):*

DNA breathing: At the end of the day, what is used in the model is the GC content on both sides of the cut site. While I recognize this has been shown to be associated with breathing, I think the authors should remain more factual about their conclusions, and stick to the observation that GC content is predictive in the abstract and introduction, writing about breathing only as a possible interpretation rather than a solid result. It would be both more precise and clearer – I struggled to understand what the paper actually showed until I reached the bottom of page 5, where the proxy for breathing is finally explained.

The model definitions are quite complex, and a cartoon of the DNA sequence, with the overhang, cut site, etc, would be very much needed to better explain the geometrical configuration of the model and the notations, as in the first Figure. It could help answer some of the following questions which confused me: which part of the sequence is subject to the PWM? Which part of the sequence is included in the sequence breathing counts on the 3' and 5' sides? Why are there 3 (and not 4, or 2) parameters in the sequence-breathing part of the model? Relatedly, why not combine distance with breathing parameters (i.e. are they redundant)? When does the 3' overhang start (we learn later it's at +2)? What is DNA shape? From what reference point is the length of deletion n counted? I'm aware there are such cartoons in the Methods, but they should be shown earlier and combined in a clearer manner, to display the definitions of the models and notations directly on the cartoon rather than painstakingly explained in the captions.

Methods: The methods are way too long for what they aim to explain: 25 pages, with a 5-page long table of notations! I would strongly recommend simplifying them to make them more readable. I will readily admit that I didn't comb through them with as much care as I would have liked, partly for lack of time, partly because I didn't feel I would learn much more from them than I already understood from the main text. I suspect very few readers will. The paper would be greatly improved by reducing its length, but also by providing key details and explanations in the Results section, to make it more self-contained.

Training set: From the methods, it appears that the Emerson dataset is first processed by IGoR to sample from the posterior distribution of scenarios. This non-obvious but essential step should be made clear in the description of the training data in the Results section.

I was also wondering why the authors didn't directly take the V-gene-dependent deletion probability distribution provided by IGoR, which (at first sight) should be strictly equivalent to sampling from that posterior distribution while being much easier. If you restrict to a single V gene, sampling scenarios and just sampling from the IGoR-provided distribution of deletion lengths for that V gene are exactly the same thing, by definition of the EM algorithm. Could you please explain that choice?

---

## [Author Response]

Essential revisions:1) A rephrasing or additional support for the claim to provide mechanistic insights which seem to be overstated. Based on the fact that only statistical models are used, there is currently no real indication of mechanistic or quantitative insight into the involved processes.

We have rephrased portions of the manuscript to emphasize that while the sequence-level features we have included in our models are mechanistically interpretable and provide quantitative statistical evidence regarding the trimming mechanism, they do not directly imply causation.

2) An extensive restructuring and rewriting of the manuscript to clarify the focus of the paper (see comments of reviewer 3, e.g. regarding the extended methods section)

As suggested by reviewer 3, we have substantially condensed and simplified the Methods section and added important methods-related details to the Results section.

3) Improved explanation of the model, as well as several details concerning the statistical analyses

We have added a figure to more clearly describe the geometrical configuration of the model features.

Reviewer #1 (Recommendations for the authors):Really great work and very interesting. My recommendations are mostly questions and some suggestions.– The text is really dense (especially when you talk about and compare all the different models tested). Can that somehow be shortened and made more concise?

We agree that the text is dense within the "Model set-up overview" section of the Results. We have created a new figure (Figure 1) to describe the geometrical configuration of the features included in each model. We hope that this cartoon will help clarify the model definitions within this section. We have also substantially shortened the main Methods section in an effort to make it more readable. Otherwise, we feel that the text detail is required to fully describe the model set-up and results.

– Maybe it's mentioned somewhere, but how many sequences were in train and test datasets? Have you performed subsampling studies to understand at what sequence number your models become stable?

This is a great suggestion! We have included a new supplementary figure (Figure 4 —figure supplement 7) showing how the magnitudes of the inferred model coefficients vary when sub-sampling the training data set. We find that the coefficients are quite stable until the size of the training data set reaches around 82,800 sequences or below. Because the full training data set used for all model training includes 21,193,153 sequences, we don't feel that our conclusions are dependent on the size of the training data set. We have added the following sentence to highlight this finding in the Results (lines 252-254):

"We noted minimal variation in the magnitude of each inferred coefficient even when changing the number of sequences included in the training data set."

We also agree that including the sequence counts for each data set would be valuable. We have now added the following sentences to the Methods section:

For the training data set (lines 515-520):

"After these processing and filtering steps, we used V-gene trimming length distributions from 21,193,153 non-productive sequences for all model training. To test each trained model, we used V-gene trimming length distributions from the remaining 107,121,841 productive sequences. From this same data set, we also used J-gene trimming length distributions from 107,255,406 productive sequences and 20,204,801 non-productive sequences to test each model."

For the independent TRA and TRB testing data sets (lines 528-534):

"From the TCRalpha data set, we used V-gene trimming length distributions from 123,496 non-productive sequences and 862,096 productive sequences and J-gene trimming length distributions from 141,451 non-productive sequences and 1,101,114 productive sequences to test each model. From the TCRbeta data set, we used V-gene trimming length distributions from 64,738 non-productive sequences and 1,435,153 productive sequences and J-gene trimming length distributions from 59,608 non-productive sequences and 1,496,953 productive sequences to test each model."

For the TRG testing data set (lines 538-541):

"We used V-gene trimming length distributions from 2,403,293 non-productive sequences and 1,002,662 productive sequences and J-gene trimming length distributions from 568,824 non-productive sequences and 250,493 productive sequences to test each model."

For the IGH testing data sets (lines 556-559):

"From these data sets, we used V-gene trimming length distributions from 160,714 non-productive sequences and 32,245 productive sequences and J-gene trimming length distributions from 297,298 non-productive sequences and 74,884 productive sequences to test each model."

– You only use unselected sequences. If you had used productive sequences, would the results have been dramatically different?

This is a good question. To explore this, we trained a new "motif + base-count-beyond" model using only productive V-gene sequences from the same cohort of individuals as the non-productive V-gene training data set and found that the inferred coefficients were highly similar between the two models. We have added a section in the Results (lines 272-281) and a Supplementary Figure (Figure 3—figure supplement 6) to highlight this. While we have used productive V-gene and J-gene sequences to test each model trained using the non-productive V-gene training data set (as shown in Figure 6—figure supplement 2), we have not explored how a model trained using productive V-gene sequences performs across non-productive sequences. Given that the inferred coefficients were highly similar between the "motif + base-count-beyond" trained using non-productive V-gene sequences and the one trained using productive V-gene sequences, we would not expect the validation results to be substantially different.

– You are somehow splitting the datasets by V gene distance. How would the results have looked like with a random train/test split?

Thanks for raising this question. We have already evaluated each model using many random held-out subsets of the V-gene training data set. To clarify this, we have changed our wording to include the word "random" when describing these held-out subsets (i.e. Figure 2 plot and caption, line 202). Overall, we found that the expected per-sequence log loss for each model was slightly higher, but quite similar, for these random subsets compared to the log loss computed across the full training data set. These results are shown in the second x-axis entry of Figure 2.

– Figure 3a, 4a what's the statistical significance between curves? I rarely see error bars (if at all) in any of the figures. To what extent are your results dependent on just sampling once?

We have provided figures 4a and 5b (formerly 3a and 4b, which we assume the reviewer is referring to since 4a was a histogram) to allow readers to qualitatively compare the model-derived trimming distributions to empirical trimming distributions (from each individual in the training data set) for the most commonly used genes. Because our model does not provide error estimates, it is not obvious to us how we would quantify statistical significance between the model-derived trimming distributions and the empirical trimming distributions for each gene.

When evaluating the expected per-sequence log loss of each model across various data sets (as in Figure 2), most of the values are computed on a single sample (e.g. a cluster of genes defined to be "most different", the full J-gene data set, etc.). However, since we have evaluated the loss of each model using many different held-out subsets of the training data set and various independent testing data sets, we feel confident that our results do not depend on just sampling once for each data set. As described in the previous suggestion, we have also calculated the loss of each model across many random, held-out subsets of the training data set. We have added a supplemental figure (Figure 2—figure supplement 1) to describe the variation in losses across the random, held-out subsets of the training data set for each model.

– You use the term mechanism a lot. The title is also quite strongly worded ("reveal role of"). To what extent is this justified given that you "only" perform statistical modeling and no experimental investigations? To what extent are you sure that your models are really a reflection of biology (causal)?

Thank you for your question. Many of the model features we have explored are inspired by previous experimental observations (i.e. certain DNA end configurations require sequence-breathing for Artemis nucleolytic action) or statistical modeling efforts (i.e. a 2x4 sequence motif is highly predictive of trimming probabilities). While our work may not directly identify mechanistic causation, the sequence-level features that we have included in our models are mechanistically interpretable and reveal quantitative statistical evidence regarding the trimming process. Additionally, because we are using statistical inference on high-throughput sequencing data from humans, in some ways, the inferences gained from our models are perhaps more a reflection of the true biological process in humans than experimental investigations using model organisms.

With that being said, we have adapted a sentence in the Discussion (lines 393-397) to highlight that our models are not causal in their design and cut back on our usage of the term "mechanism" throughout the manuscript (i.e. see lines 113, 194, etc.) We have also further emphasized that our models serve to quantify the effects of various sequence-level features in the trimming process rather than provide mechanistic causal evidence (see lines 232-234, 400, 438-439, etc.).

Reviewer #2 (Recommendations for the authors):DNA breathing: At the end of the day, what is used in the model is the GC content on both sides of the cut site. While I recognize this has been shown to be associated with breathing, I think the authors should remain more factual about their conclusions, and stick to the observation that GC content is predictive in the abstract and introduction, writing about breathing only as a possible interpretation rather than a solid result. It would be both more precise and clearer – I struggled to understand what the paper actually showed until I reached the bottom of page 5, where the proxy for breathing is finally explained.

This is a good suggestion and we have changed our wording of the model terms from "the capacity for sequence-breathing" to "the GC nucleotide content in both directions of the wider sequence" throughout the manuscript (for example, see lines 21-26 and 118-124). For clarity, we now describe sequence-breathing as a possible interpretation of the GC nucleotide content model terms within the Abstract (lines 23-25) and Introduction (lines 118-120). We also added a sentence highlighting the previously-identified role of sequence-breathing in the trimming mechanism within the Introduction (lines 88-90) to add justification to our interpretations. We aren't aware of any other nucleotide-level determinants of sequence-breathing, but we're happy to hear of other approaches.

The model definitions are quite complex, and a cartoon of the DNA sequence, with the overhang, cut site, etc, would be very much needed to better explain the geometrical configuration of the model and the notations, as in the first Figure. It could help answer some of the following questions which confused me: which part of the sequence is subject to the PWM? Which part of the sequence is included in the sequence breathing counts on the 3' and 5' sides? Why are there 3 (and not 4, or 2) parameters in the sequence-breathing part of the model? Relatedly, why not combine distance with breathing parameters (i.e. are they redundant)? When does the 3' overhang start (we learn later it's at +2)? What is DNA shape? From what reference point is the length of deletion n counted? I'm aware there are such cartoons in the Methods, but they should be shown earlier and combined in a clearer manner, to display the definitions of the models and notations directly on the cartoon rather than painstakingly explained in the captions.

Thank you for this great suggestion! We have included a new figure (Figure 1) to describe the geometrical configuration of an example V-gene sequence and the features included in each model. We hope that this cartoon will answer most of the questions you have listed.

Methods: The methods are way too long for what they aim to explain: 25 pages, with a 5-page long table of notations! I would strongly recommend simplifying them to make them more readable. I will readily admit that I didn't comb through them with as much care as I would have liked, partly for lack of time, partly because I didn't feel I would learn much more from them than I already understood from the main text. I suspect very few readers will. The paper would be greatly improved by reducing its length, but also by providing key details and explanations in the Results section, to make it more self-contained.

We have reduced the length of the main Methods section to 7.5 pages and included additional details (i.e. the notation table) as appendices. We do feel that the full detail is important for readers wishing to fully understand and reproduce the results.

We have also moved some key methods-related details to the Results section (for example, IGoR details on lines 132-138). We hope that these changes will make the Methods more readable.

Training set: From the methods, it appears that the Emerson dataset is first processed by IGoR to sample from the posterior distribution of scenarios. This non-obvious but essential step should be made clear in the description of the training data in the Results section.

We have added the following sentences (lines 132-138 in the manuscript) to the Results section to clarify the methods used to annotate each sequence in the training data set:

"V(D)J recombination scenarios were assigned to each sequence for each individual using the IGoR software which can learn unbiased V(D)J recombination statistics from immune sequence reads (Marcou et. al, Nature Communications 2018). Using these V(D)J recombination statistics, IGoR output a list of potential recombination scenarios with their corresponding likelihoods for each TCRb-chain sequence in the training data set. We annotated each sequence with a single V(D)J recombination scenario by sampling from these potential scenarios according to the posterior probability of each scenario (see Methods for further details)."

I was also wondering why the authors didn't directly take the V-gene-dependent deletion probability distribution provided by IGoR, which (at first sight) should be strictly equivalent to sampling from that posterior distribution while being much easier. If you restrict to a single V gene, sampling scenarios and just sampling from the IGoR-provided distribution of deletion lengths for that V gene are exactly the same thing, by definition of the EM algorithm. Could you please explain that choice?

This is a great point! We chose the posterior-sampling approach for several reasons. First, this approach allowed us to consider all possible V-gene (and J-gene) annotations for each sequence. Also, because we were interested in training each model using V-gene sequences from the Emerson data set and validating each model using J-gene sequences from the same data set, sampling scenarios from the posterior distribution allowed us to simultaneously infer V-gene and J-gene trimming amounts in a single sequence annotation step. Lastly, while we began our analyses by annotating each sequence with the "most-parsimonious" scenario, to avoid possible biases introduced by that annotation approach, we chose to transition to a probabilistic annotation approach (using IGoR). Because our modeling pipeline was already built to intake a complete sequence annotation (including gene choices, trimming amounts, and insertions), we chose to sample scenarios from the posterior distribution produced by IGoR which outputs a similar data structure.